# Anderson critical metal phase in trivial states protected by average magnetic crystalline symmetry

Fa-Jie Wang[1], Zhen-Yu Xiao[1], Raquel Queiroz [2], B. Andrei Bernevig [3], Ady Stern [4] & Zhi-Da Song[1,5,6] ✉

Transitions between distinct obstructed atomic insulators (OAIs) protected by crystalline symmetries, where electrons form molecular orbitals centering away from the atom positions, must go through an intermediate metallic phase. In this work, we find that the intermediate metals will become a scale-invariant critical metal phase (CMP) under certain types of quenched disorder that respect the magnetic crystalline symmetries on average. We explicitly construct models respecting average $C_{2z}T$, $m$, and $C_{4z}T$ and show their scale-invariance under chemical potential disorder by the finite-size scaling method. Conventional theories, such as weak anti-localization and topological phase transition, cannot explain the underlying mechanism. A quantitative mapping between lattice and network models shows that the CMP can be understood through a semi-classical percolation problem. Ultimately, we systematically classify all the OAI transitions protected by (magnetic) groups $Pm$,$P2'$,$P4'$, and $P6'$ with and without spin-orbit coupling, most of which can support CMP.

The interplay between topology and the Anderson (de)localization has provided an understanding of the quantum Hall transition[1,2] and the classification (not including crystalline symmetries)[3–5] of topological insulators (TIs)[6–9]. A remarkable result of this interplay is the delocalization in TIs protected by local symmetries[3,5]: In the presence of disorder that does not induce a bulk phase transition, the TI surface states[10–13] are guaranteed to be delocalized; In the bulk, a disorder-induced transition between trivial and topological insulators must go through a divergent localization length[14,15]. Similar behavior occurs also in topological phases that require crystalline symmetry to be preserved, despite the breaking of that symmetry by disorders, as long as the symmetry is preserved on average. Examples include weak topological insulators[14,16–22] and some topological crystalline insulators (TCIs)[23–35] (including higher-order states with hinge modes[36–42]). For instance, inversion-symmetry-protected axion insulators[26,43–45] have hinge modes[46], and their phase transitions to trivial insulators must experience a delocalized diffusive metal phase if the disorder respects an average inversion symmetry[47,48]. Here, as defined in refs. 14,49, average symmetry is the symmetry of an ensemble comprising different disorder realizations on a local Hamiltonian. The average symmetry operation transforms an individual system into another with the same realization probability. Also, we require each system in the ensemble to be self-average. Even though not mathematically proven, most TCIs with protected boundary states are believed to be stable against disorders respecting the crystalline symmetries on average. This can be understood intuitively: Suppose the disorder potential slowly varies in real space. Then, during the transition, the disordered system can be divided into topological and trivial regions. Boundary states between the two types of regions must exist as promised by stable topology, giving rise to the delocalized phase transition.

In this work, we find that such delocalization behavior also generalizes to some topologically trivial states. There are two types of non-atomic states that are not conventional TIs or TCIs−the fragile

¹International Center for Quantum Materials, School of Physics, Peking University, 100871 Beijing, China. ²Department of Physics, Columbia University, New York, NY, USA. ³Department of Physics, Princeton University, Princeton, NJ 08544, USA. ⁴Department of Condensed Matter Physics, Weizmann Institute of Science, Rehovot 7610001, Israel. ⁵Hefei National Laboratory, Hefei 230088, China. ⁶Collaborative Innovation Center of Quantum Matter, Beijing 100871, China. ✉e-mail: songzd@pku.edu.cn

topological insulators[50–53] and the obstructed atomic insulators (OAIs)[30,54–56]. The former has a Wannier obstruction that can be removed by adding trivial bands and was recently found to hold delocalized critical states[57]. However, the latter is completely trivial and can be wannierized to molecular orbitals with charge centers away from the atoms. Given OAIs' localized nature, it would be surprising if they can have delocalized states in the presence of disorder. In this work, we demonstrate that such delocalization does exist and is actually a common feature in transitions between magnetic OAIs. Conventional scenarios for delocalization theories, such as weak antilocalization (requiring time-reversal symmetry) and topological phase transition (requiring stable boundary states) cannot explain the underlying mechanism behind the delocalized states that we find.

## Results

### A $C_{2z}T$-symmetric quantum network model

We first investigate several OAIs protected by the $C_{2z}T$ symmetry without time-reversal symmetry (TRS)[58–61]. Later, we will generalize the discussion to other magnetic point groups in subsection "OAI transitions in generic magnetic point groups". The OAIs belong to the Altland-Zirnbauer symmetry class A[62], where all states except the quantum Hall transition point were expected to be localized[63]. These OAIs are characterized by the $\mathbb{Z}_2$ Real Space Invariant (RSI) $\delta_w$, which is protected by $C_{2z}T$[64,65] and takes the value $\delta_w = 1$ if the associated $C_{2z}T$ center $w$ is occupied by an odd number of Wannier functions and zero otherwise. When the $\delta_w$'s of a system have the same value at all the $C_{2z}T$ centers, they are equivalent to the second Stiefel-Whitney class $w_2$[60,66].

Two features of a $w_2 = 1$ insulator are worth emphasizing here: First, it can be regarded as a $C_{2z}T$-protected fragile insulator plus two additional trivializing bands. And second, to tune a $w_2 = 1$ insulator into a $w_2 = 0$ insulator, one has to first close the gap by creating pairs of Dirac points, which are locally stabilized by the $C_{2z}T$ symmetry, then braid[60,67,68] these Dirac points with other Dirac points inside the valence bands, and only then reopen a gap by annihilating the Dirac points (see subsection "Regularizing the network model to lattice models" below). As a consequence, when the transition is driven by the variation of one parameter, there is a gapless transition region rather than a transition point. Using a finite-size scaling procedure, we find that this gapless region becomes a critical metal phase (CMP) when disorder is added, provided that the disorder respects $C_{2z}T$ on average. Electronic states in CMP are delocalized and contribute to a scale-invariant conductance in the thermodynamic limit. We also find CMPs with other average symmetries and RSIs, suggesting CMP is a common feature of magnetic OAI transitions.

CMP has been numerically observed in systems with random fluxes[69–71] or random spin-orbit coupling combined with a magnetic field[72]. Inspired by these works, we first argue the existence of $C_{2z}T$-stabilized CMP through a semi-classical percolation theory. We then relate the percolation theory to a quantum network model[73] and further map it to lattice models for the $C_{2z}T$-protected OAIs.

To present a scenario that naturally leads to a CMP, we consider a system that is tessellated by three types of randomly sized and shaped insulating regions, whose Chern numbers are 0, 1 and −1, respectively, as shown in Fig. 1a, b. Physically, the fluctuation of Chern numbers could arise from random fluxes. The area fraction of Chern number $C$ is $p_C$. By definition $\sum_C p_C = 1$. Since the operation $C_{2z}T$ reverses the sign of Chern numbers, an average $C_{2z}T$ symmetry means $p_1 = p_{-1}$. Then there are two distinct phases. If $p_0 > \frac{1}{2}$, according to the classical percolation theory[74], the $C = 0$ regions form an extensive cluster while the $C = \pm 1$ regions form isolated islands and can be continuously shrunk to zero. Thus, $p_0 > \frac{1}{2}$ should correspond to a localized phase (LP). If $p_0 < \frac{1}{2}$, it is instead the $C = 0$ regions that can be shrunk to zero, and the system is equivalently tessellated by the

$C = \pm 1$ regions with the same area fraction $\frac{1}{2}$. As in the quantum Hall transition[73], the chiral edge states between the $C = \pm 1$ regions connect to an extensive cluster with a fractal dimension and contribute to a scale-invariant conductance in the thermodynamic limit. Thus, $p_0 < \frac{1}{2}$ should correspond to the CMP.

We can simulate the above percolation problem with a quantum network model on the Manhattan lattice[75] (Fig. 1c). The red and blue squares are Chern blocks with $C = 1$, −1, respectively. The chiral edge modes between them and the trivial (white) regions form horizontal and vertical wires, and at each intersection, an electron can go straight or turn either left or right, depending on the type of intersection. The scattering equation at one intersection reads

$$\begin{pmatrix} \psi_3 \\ \psi_4 \end{pmatrix} = \begin{pmatrix} \cos\theta & -i\sin\theta \\ -i\sin\theta & \cos\theta \end{pmatrix} \begin{pmatrix} \psi_1 \\ \psi_2 \end{pmatrix}, \tag{1}$$

where $\psi_{3,4}$ and $\psi_{1,2}$ are the outgoing and incoming modes, respectively, and $\theta$ is the single parameter that determines the probability amplitudes of going straight ($\cos\theta$) and turning left or right ($-i\sin\theta$). The model in the clean limit has symmetries of the magnetic space group $P_C4bm$ (#100.177 in BNS setting) generated by $C_{2z}T$, $C_{4z}$, and $m_{xy}$ symmetries[76]. The symmetry elements of the generators are shown in Fig. 1c. (Notice that $C_{2z}T$ centers do not coincide with $C_{4z}$ centers, and $C_{4z}^2 \cdot C_{2z}T$ is a magnetic translation, i.e., a translation followed by time reversal. See Supplementary Information for more details.) One can see that the $C_{2z}T$ operation interchanges the $C = \pm 1$ regions. Note that the Manhattan lattice is not the only way to simulate the percolation problem. A Kagome-like network also works, with a localization behavior that is similar to the Manhattan lattice (See Section II.A of Supplementary Information).

Random sizes of the Chern blocks are simulated by the random propagation phases, or, equivalently, random vector potentials, along the bonds between intersections. When $\theta = \pm\frac{\pi}{2}$, the chiral modes form local current loops surrounding $C = \pm 1$ regions, and the $C = 0$ regions are effectively connected. When $\theta \to 0$, the chiral modes are almost decoupled wires, and $C = 0$ regions are effectively separated. Thus, $\theta = \pm\frac{\pi}{2}$ and $\theta \to 0$ should correspond to the localized limits ($p_0 = 1$) and the CMP limit ($p_0 = 0$), respectively. According to the percolation argument, there will be a critical value $\theta_c$ below (above) which the system enters the critical (localized) phase.

We use the transfer matrix techniques[77] of quasi-1D systems, where longitudinal size $M$ is much larger than transversal size $L$, to calculate the quasi-1D localization length $\rho$ for finite $L$'s. More on this is summarized in the Method section, and one can read Section V of Supplementary Information for the full technical details. The normalized quasi-1D localization length $\Lambda = \rho/L$ is an indicator of the (de) localization: Divergent, finite, and vanishing $\Lambda$'s in the limit $L \to \infty$ indicate metallic, critical, and localized states, respectively. As shown in Fig. 1e, $\Lambda$ decreases with $L$ for $|\theta| > \theta_c \approx \frac{\pi}{4}$ and is almost independent of $L$ for $|\theta| < \theta_c$. Hence, $|\theta| \in (\theta_c, \pi/2]$ and $|\theta| \in (0, \theta_c)$ correspond to LP and CMP, respectively, which confirms the percolation argument. Note that error bars in plots of this work represent the standard deviations (SD) of corresponding data points. The conductance is also calculated and shown in Fig. 1f. The $\beta$-function $\beta = d\ln G/d\ln L$ derived from the conductance data vanishes in the thermodynamics limit above some critical conductance $G_c = 2 \sim 3e^2/h$ (see subsection "$\beta$-function of the CMP" for details). The behavior of $\beta$-function further establishes the criticality of CMP and may suggest that the CMP-LP transition is similar to the Berezinskii-Kosterlitz-Thouless transition[78,79].

We can define the network model through its Hamiltonian $H_N$ rather than through the scattering matrix Eq. (1). The Hamiltonian has only three parameters, velocity $v$ of the chiral modes, the $\delta$-potential $\lambda$

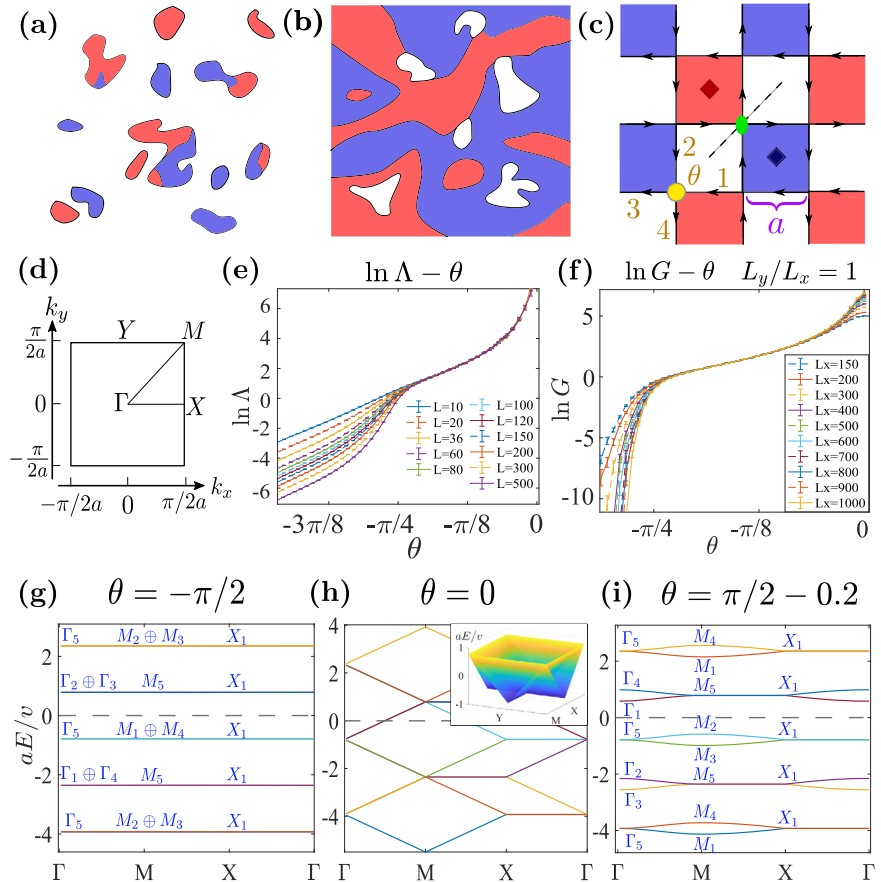

**Fig. 1 | The network model $H_N$. a, b** Percolation systems with $p_0 > 1/2$ and $p_0 < 1/2$. **c** The network model on the Manhattan lattice, where the dashed line, dark red and blue rhombuses, and green oval indicate the $M_{xy}$ mirror plane, $C_{4z}$ centers and $C_{2z}T$ center, respectively. At every intersection (e.g. yellow circle), there is a scattering potential $\lambda$, and the scattering angle $\theta$ is determined by $\lambda = \theta v$ with $v$ being the velocity of chiral modes. **d** Brillouin zone and high symmetry points. **e** Normalized quasi-1D localization length $\Lambda$'s as functions of $\theta$ at different transversal system sizes $L$. The used longitudinal system size is $M = 10^7$ and the precision ($\sigma_N/\Lambda$) has

reached 1%. (For network model, 'size' refers to the number of squares.) $\Lambda$ only depends on $|\theta|$, hence only data with $\theta < 0$ is shown. **f** The mean conductances (over $10^3$ square-shaped samples) as the function of $\theta$ and the precision ($\sigma_G/G$) has reached 0.5% in the delocalized phase. **g–i** Band structures of the network model at $\theta = -\pi/2$, $0$, $\pi/2 - 0.2$. Blue capital letters indicate the associated irreps. The inset in **h** is the 3D plot of the dispersion of the middle two bands around the zero energy indicated by the dashed lines.

**Table 1 | Character table of irreps at high symmetry momenta in magnetic space group $P_C4bm$ (#100.177 in BNS setting), taken from the COREPRESENTATIONS program on the Bilbao Crystallographic Server[80]**

|  | $\Gamma_1$ | $\Gamma_2$ | $\Gamma_3$ | $\Gamma_4$ | $\Gamma_5$ |  | $M_1$ | $M_2$ | $M_3$ | $M_4$ | $M_5$ |  | $X_1$ |
|---|---|---|---|---|---|---|---|---|---|---|---|---|---|
| $\{1\|0,0,0\}$ | 1 | 1 | 1 | 1 | 2 | $\{1\|0,0,0\}$ | 1 | 1 | 1 | 1 | 2 | $\{1\|0,0,0\}$ | 2 |
| $C_{2z} = \{2_{001}\|-\frac{1}{2},\frac{1}{2},0\}$ | 1 | 1 | 1 | 1 | −2 | $C_{2z} = \{2_{001}\|-\frac{1}{2},\frac{1}{2},0\}$ | −1 | −1 | −1 | −1 | 2 | $C_{2z} = \{2_{001}\|-\frac{1}{2},\frac{1}{2},0\}$ | 0 |
| $C_{4z} = \{4^+_{001}\|0,\frac{1}{2},0\}$ | 1 | −1 | −1 | 1 | 0 | $C_{4z} = \{4^+_{001}\|0,\frac{1}{2},0\}$ | $i$ | $-i$ | $-i$ | $i$ | 0 | $\{m_{100}\|0,\frac{1}{2},0\}$ | 0 |
| $M_{xy} = \{m_{1\bar{1}0}\|0,0,0\}$ | 1 | −1 | 1 | −1 | 0 | $M_{xy} = \{m_{1\bar{1}0}\|0,0,0\}$ | −1 | 1 | −1 | 1 | 0 | $\{m_{010}\|\frac{1}{2},0,0\}$ | 0 |

Characters of the listed symmetry operations can uniquely determine the irreps One should notice that we use a different convention of the origin point as the Bilbao Crystallographic Server. Our $C_{2z} = \{2_{001}\|-\frac{1}{2},\frac{1}{2},0\}, C_{4z} = \{4^+_{001}\|0,\frac{1}{2},0\}, M_{xy} = \{m_{1\bar{1}0}\|0,0,0\}, \{m_{100}\|0,\frac{1}{2},0\}$, and $\{m_{010}\|\frac{1}{2},0,0\}$ correspond to $\{2_{001}\|0,0,0\}, \{4^+_{001}\|0,0,0\}, \{m_{1\bar{1}0}\|\frac{1}{2},-\frac{1}{2},0\}, \{m_{100}\|\frac{1}{2},\frac{1}{2},0\}$, and $\{m_{010}\|\frac{1}{2},-\frac{1}{2},0\}$ in the standard convention of the Bilbao Crystallographic Server, respectively.

at each intersection, and the lattice constant $2a$.

$$H_N = \sum_{d,l} i(-1)^l v \int d\xi \, \psi^\dagger_{d,l}(\xi) \partial_\xi \psi_{d,l}(\xi) \\ + \sum_{ll'} \lambda [\psi^\dagger_{h,l}(l'a)\psi_{v,l'}(la) + h.c.] \quad (2)$$

The subscript $d = v$, h represents the vertical or horizontal orientations of the wires. $l = 0, \pm 1 \cdots$ distinguishes different parallel wires. $\xi$ is the coordinate inside one wire. The operators $\psi_{v,l}(\xi)$ and $\psi_{h,l}(\xi)$ act at the

real space positions $(x, y) = (la, \xi)$ and $(\xi, la)$, respectively. As established in Section III.B of Supplementary Information, the scattering angle $\theta$ is determined by the Hamiltonian parameters as $\theta = \lambda/v$. The evolution of the band structure as $\theta$ changes from the localized limit $\theta = -\pi/2$ to the other localized limit $\theta = \pi/2$ is illustrated in Fig. 1g–i. At $\theta = 0$, the network model decouples to vertical and horizontal chiral wires, and the corresponding dispersion becomes quasi-1D (Fig. 1h). The symbols appearing in the figure, $\Gamma_n, M_n$ $(n = 1\cdots 5)$ and $X_1$, represent irreducible representations (irreps) of $P_C4bm$ and are defined in Table 1. It is worth mentioning that because the chiral modes have

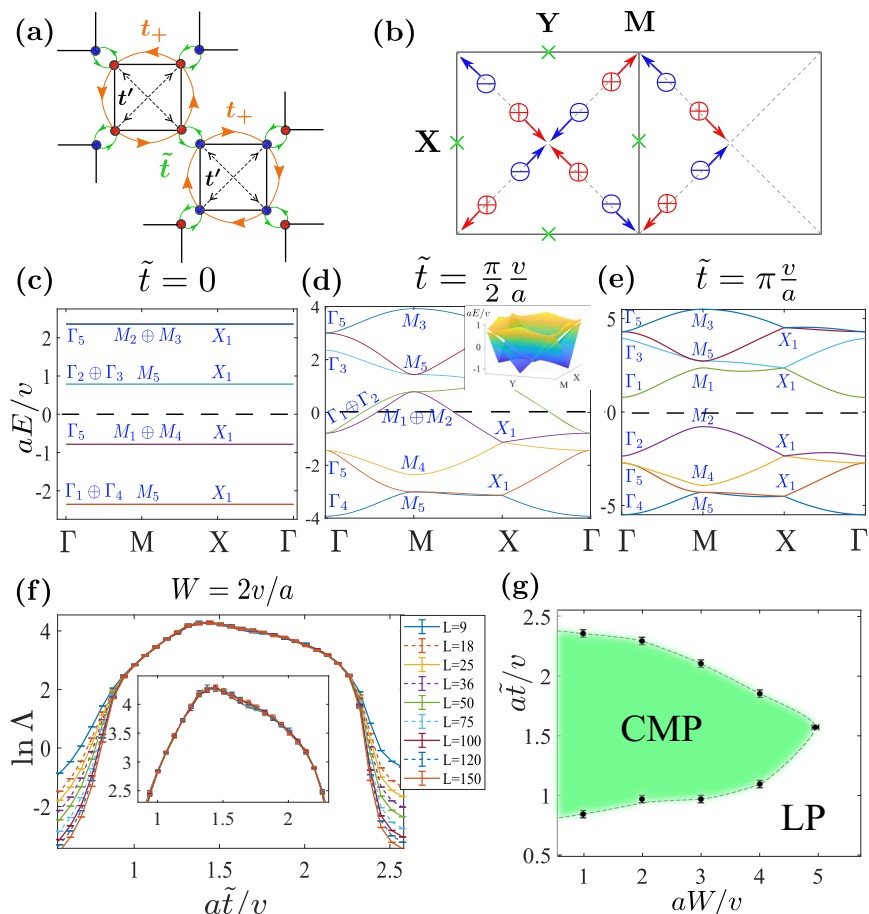

**Fig. 2 | The lattice model $H_{8B}$. a** The unit cell and hoppings of $H_{8B}$. **b** The evolution of Dirac points between the fourth and fifth bands. Red (blue) circles indicate Dirac points with positive (negative) chirality (defined in Section IV.D of Supplementary Information), and the green crosses correspond to Dirac points between the third and fourth bands. The blue and red arrows indicate the evolution directions of Dirac points when $\tilde{t}$ increases. **c**–**e** Band structures with $a\tilde{t}/v = 0, \pi/2, \pi$ (correspond to $\theta = -\pi/2, 0, \pi/2$, respectively). The blue capital letters indicate the irreps at high

symmetry points. The dashed lines indicate zero energy. **f** The normalized localization length as functions of $\tilde{t}$ at various transversal sizes $L$ and fixed disorder strength $W = 2v/a$ and Fermi level $E_F = 0$. The used longitudinal size is $M = 10^7$ and the precision ($\sigma_\Lambda/\Lambda$) has reached 3%. (For lattice models, `size' refers to the number of unit cells.) **g** The phase diagram in $\tilde{t} - W$ plane for $E_F = 0$, where the green region represents the CMP enclosed by the LP.

---

unbounded energies, this Hamiltonian has an infinite number of bands that are periodic in energy, i.e., $E_n(\mathbf{k}) = E_{n+8}(\mathbf{k}) + 2\pi v/a$. For example, the lowest branch (two connected bands) in Fig. 1g–i is identical to the highest branch. Tracing the evolution (detailed in Supplementary Fig. 18), we find the phase transition process between these two LPs can be depicted by the irrep exchange at the zero energy (dashed lines in Fig. 1g–i)

$$-\Gamma_1 - M_1 + \Gamma_2 + M_2, \tag{3}$$

where a minus (plus) sign means an energy level with the associated irrep crosses the zero energy from below (above) to above (below) during the phase transition. One may notice that the occupied $\Gamma_4$ state in Fig. 1g also changes into $\Gamma_3$ in Fig. 1i. However, this change is realized by level exchanges between the lowest branch in the figures and the lower branches beyond the scope of the figures. Since $\Gamma_{3,4}$ do not cross the zero energy, they are not counted in the phase transition. We now relate the phase transition (3) to a change in the index $\delta_w$.

**Regularizing the network model to lattice models**

In order to see the band topology, we need to regularize the network model to a lattice model. Our strategy is to use the local current loop states (flat bands in Fig. 1g) in the localized limit $\theta = -\frac{\pi}{2}$ as a basis set and then truncate the basis according to their energies. The minimal

model that can reproduce the phase transition in Eq. (3) is constructed from the upper eight consecutive flat bands in Fig. 1g to obtain the Hamiltonian $H_{8B}$ shown in Fig. 2a. (See Section IV.C of Supplementary Information for more details.) It has eight orbitals that are respectively located at the eight corners of the two squares in one unit cell, which correspond to the two Chern blocks in Fig. 1c. The explicit form of $H_{8B}$ can be expressed as

$$H_{8B} = \tilde{t} \sum_{\langle p,q \rangle} c_p^\dagger c_q + t \sum_{\langle\langle p,q \rangle\rangle} e^{i\phi_{pq}} c_p^\dagger c_q + t' \sum_{\langle\langle\langle p,q \rangle\rangle\rangle} c_p^\dagger c_q, \tag{4}$$

where $p, q$ are the site indices, $\langle \cdot \rangle$, $\langle\langle \cdot \rangle\rangle$, and $\langle\langle\langle \cdot \rangle\rangle\rangle$ represent the green (nearest neighbor), orange (square edges), and dashed black (square diagonals) bonds in Fig. 2a, respectively. The hopping parameters are given by $\tilde{t} = (\theta + \frac{\pi}{2})\frac{v}{a}, t = \frac{\sqrt{2}\pi v}{4a}, t' = -\frac{\pi v}{4a}$. The phase factor $\phi_{pq}$ equals $\frac{3}{4}\pi$ ($-\frac{3}{4}\pi$) if the associated hopping is parallel (anti-parallel) to the orange arrows, which are clockwise and anticlockwise for squares formed by the blue and red sites, respectively. For simplicity, we also denote the complex hopping $te^{\pm i\frac{3}{4}\pi}$ as $t_\pm$ in the following.

We take the Fermi level to be at $E_F = 0$ and focus on that energy. The Hamiltonian $H_{8B}$ reproduces the flat bands in the localized limit $\theta = -\frac{\pi}{2}$ when $\tilde{t} = 0$, where blue and red squares are decoupled from each other (Fig. 2c). The four flat valence bands are molecular orbitals

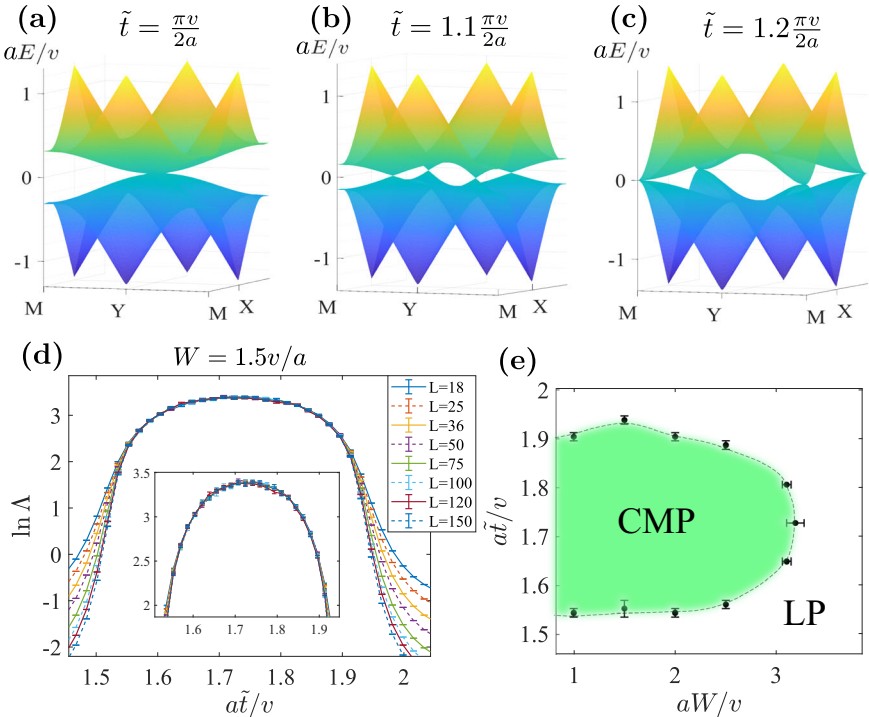

**Fig. 3 | The lattice model $H'_{8B}$. a–c** Evolution of the middle two bands with $\tilde{t}$. **d** Normalized localization length $\Lambda$ as functions of $\tilde{t}$ with $W = 1.5v/a$, $E_F = 0$ and various transversal sizes $L$. The used longitudinal size is $M = 10^7$ and the precision $(\sigma_\Lambda/\Lambda)$ has reached 2%. **e** The phase diagram in $\tilde{t} - W$ plane with $E_F = 0$, where the green region represents the critical metal phase enclosed by the localized phase. Note that $H'_{8B}$ in the clean limit has an additional chiral symmetry which fixes the Dirac points at zero energy. However, the chiral symmetry is broken by the chemical potential disorder and is not responsible for CMP (See Supplementary Fig. 10 and Section IV.E for details).

at the square centers ($C_{4z}$ centers). Since they do not occupy the $C_{2z}T$ centers (squares corners), all the corresponding RSIs $\delta_w = 0$ and the Stiefel-Whitney class $w_2 = 0$. As $\tilde{t}$ increases, $H_{8B}$ closes its gap, and when $\tilde{t} = \frac{\pi v}{2a}$, it reproduces the quasi-1D bands of the network in the decoupled wire limit $\theta = 0$ except for small deviations (non-linear dispersions) as if the wires are weakly coupled. As a consequence of the $C_4$ symmetry, the Dirac points between the gapless bands occur at the same energy. (See Figs. 1h and 2d) As $\tilde{t}$ continues to increase, $H_{8B}$ reopens a gap (Fig. 2e), and this gap continues to the limit $\tilde{t} \to \infty$. By tracing the evolution of the energy levels, one can verify that the irrep exchange at $E = 0$ is indeed the same as Eq. (3). In the limit $\tilde{t} \to \infty$, electrons form bonding states at the four $C_{2z}T$ centers (per cell), i.e., green bonds in Fig. 2a. Thus, the final state has $\delta_w = 1$ at all $C_{2z}T$ centers, i.e., $w_2 = 1$. As detailed in the Method section, the phase transition can be further confirmed through the machinery of Topological Quantum Chemistry[30,80].

As shown in Fig. 2d, (tilted) Dirac points between the fourth and fifth bands are created in the phase transition process. The evolution of Dirac points is sketched in Fig. 2b, where the trajectory forms a closed path enclosing the underlying Dirac point at $X$ between the third and fourth bands[60,67,68]. A more detailed discussion is given in Section IV.D of Supplementary Information.

It is worth mentioning that the vector potential disorder we used in the network model is now mapped to a chemical potential disorder in $H_{8B}$ (plus two times weaker hopping disorders that will be omitted). We leave the mathematical analysis in Section IV.F of Supplementary Information and only present a heuristic argument here. The basis of $H_{8B}$ (Fig. 2a) can be thought as wave-packets of the chiral modes that simultaneously have position centers and momentum centers. The position centers, by construction, are located at the square corners. We denote their 1D momentum centers as $q_c$. Then a vector potential $A$ will shift a momentum center $q_c$ to $q_c + A$ and result in an energy shift $vA$. Therefore, the resulting disorder potential in $H_{8B}$ should have a

large on-site component. We also ignore the correlations among on-site random potentials for simplicity and efficiency. In numerical calculations, we only use uncorrelated on-site disorder and choose the disorder potential equally distributed in $[-W/2, W/2]$. Test calculations with full projected disorder potentials and correlations show no qualitative difference (see Supplementary Fig. 30).

We calculated the normalized localization length $\Lambda$ as a function of $\tilde{t}$ with $E_F = 0$ and $W = 2v/a$ (Fig. 2f). The system is localized when $|\tilde{t} - \pi v/2a| > \Delta_c \approx 0.6v/a$, corresponding to the two OAI limits, and becomes critical when $|\tilde{t} - \pi v/2a| < \Delta_c$. The criticality has been examined for large transversal sizes up to $L = 500$ unit cells (2000 atoms). We also calculate $\Lambda$ at other $\tilde{t}$'s and $W$'s. From these data, we can determine a phase diagram shown in Fig. 2g, where the dashed line separates the CMP inside and the LP outside it. (See Supplementary Fig. 8 for details of large-scale examination and phase boundary determination.) Since $H_{8B}$ does not have chiral or particle-hole symmetries, the choice $E_F = 0$ is not special in terms of symmetries. We have confirmed that CMP also exists when $E_F \neq 0$ as long as the OAI limits are intact (See Supplementary Fig. 9).

For the CMP in Fig. 2g, if we turn off the disorder, the resulting clean system has a finite density of states (DOS) around the zero energy, which may lead to a large localization length that may exceed the numerically accessible transversal size. To rule out possible finite-size effects, we consider a lattice model $H'_{8B}$ that has the same crystalline symmetries and topology as $H_{8B}$ but vanishing DOS at the zero energy. Such a $H'_{8B}$ can be obtained from $H_{8B}$ by (i) removing diagonal hopping $t'$ and (ii) changing the edge hopping to $t_\pm = (\pm Ai - 1)t$, where $A$, chosen as 1.2 hereafter, is an extra parameter that controls the range of the critical phase. (See Section IV.E of Supplementary Information for more details.) The hopping $\tilde{t}$ (green bond in Fig. 2a) at $C_{2z}T$ center remains unchanged. The $\tilde{t} = 0$ and $\tilde{t} \to \infty$ limits still represent the two OAI limits with charge centers at the $C_{4z}$ and $C_{2z}T$ centers, respectively. Hence, changing $\tilde{t} = 0$ to $\tilde{t} = \infty$ will change $w_2 = 0$ to $w_2 = 1$ for the lower

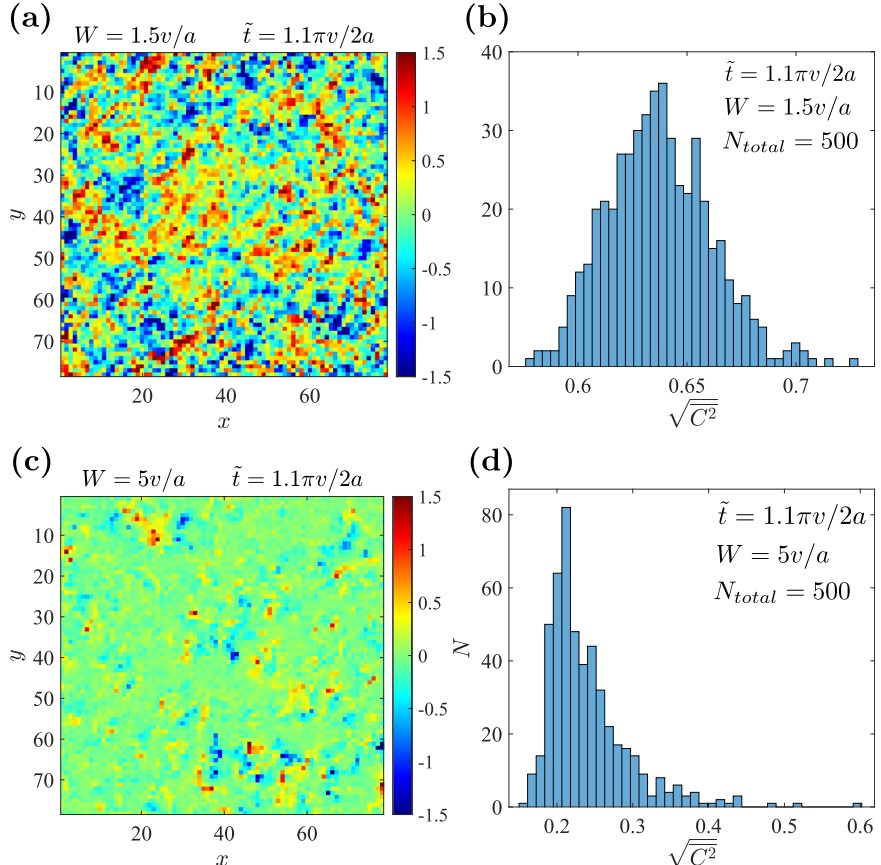

**Fig. 4 | Typical topographies and statistics of LCMs in CMP and LP. a** Typical topography (80 × 80 cells) of LCMs of $H'_{8B}$ in the CMP ($\tilde{t}=1.1\pi v/2a, W=1.5v/a$). **b** Distribution of $\sqrt{\langle C^2 \rangle}$ in 500 disorder configurations with the same parameters

as **a**. **c** and **d** are the same plots as **a** and **b** but in the localized phase with $\tilde{t}=1.1\pi v/2a, W=5v/a$.

four bands as it did in $H_{8B}$. Fig. 3a–c depict the evolution of the fourth and fifth bands, where the gap only closes at the four (untilted) Dirac points, resulting in a zero DOS at the zero energy.

Parallel to the results of $H_{8B}$ shown in Fig. 2f, g, we show Λ with $E_F=0$ and fixed $W$ and the phase diagram for $H'_{8B}$ in Fig. 3d, e, respectively, where no qualitative difference for the CMP is found. Thus, the potential finite-size effect of $H_{8B}$ due to large DOS is ruled out. Additionally, we did calculations with $E_F \neq 0$ and obtained similar results as $E_F = 0$ (See Supplementary Fig. 10).

## Local Chern markers

In order to verify the percolation argument directly, we refer to a widely used local topological marker called local Chern marker (LCM)[81–83], which reformulates the Chern number locally in real space without summing over the whole sample. The LCM of one unit cell is

$$C(\mathbf{R}) = \frac{4\pi}{A_c} \sum_{\alpha} \mathrm{Im}\langle \mathbf{R}\alpha | \hat{P}\hat{x}\hat{P}\hat{y}\hat{P} | \mathbf{R}\alpha \rangle \tag{5}$$

where $\mathbf{R}$ is the position of the unit cell, $\alpha$ indicates the orbitals inside one unit cell, $A_c$ is the area of one unit cell, and $\hat{P}$ is the projection operator of the occupied states.

Inside a macroscopic region where a gap is well preserved, the LCM converges to the quantized Chern number, whereas around gapless regions such as the boundaries, LCM may strongly fluctuate[82]. In our models, due to the percolation argument, regions with a single Chern number $C$ ($=\pm 1$) never become extended because the fraction $p_C$, as required by the $C_{2z}T$, is always smaller than $\frac{1}{2}$ for $p_0 > 0$.

Therefore, no macroscopic Chern block is expected for a general point in the CMP. Nevertheless, the microscopically inhomogeneous LCM can still reflect the local topological properties and can be applied to, for example, disordered systems near topological phase transitions[84]. Figure 4a and c show the topography of LCM of $H'_{8B}$ in the CMP and LP with given disorder configurations, respectively. In the CMP, the sample is dominated by randomly distributed positive and negative Chern cells, consistent with the percolation argument that $C=\pm 1$ regions together are extended through the whole system. In the LP, the LCM almost vanishes everywhere. We also calculate the distribution of the second moment of LCM $\sqrt{\langle C^2 \rangle}$ over 500 disorder configurations in both the CMP and LP (Fig. 4b, d), where $\langle \cdot \rangle$ means averaging over all the cells. In the CMP (LP), $\sqrt{\langle C^2 \rangle} > \frac{1}{2}$ ($< \frac{1}{2}$) for most configurations, i.e., the regions with non-zero (zero) Chern numbers dominate. These phenomena confirm our semi-classical percolation picture.

## OAI transitions in generic magnetic point groups

For a qualitative understanding of the CMP, we note that a local breaking of $C_{2z}T$ symmetry allows for a local gap of the Dirac nodes. This gap makes the region of the Dirac point carry a spread Berry curvature that integrates to $\pm \pi$, with the sign being determined by the chirality of the Dirac point multiplied by the sign of the gaping mass. In the cases we considered here, all Dirac points occur at the same energy. Denoting the number of Dirac points by $2N$, there are $2^{2N}$ assignments of the signs of the gaping mass, out of which $(2N)!/(N!)^2$ lead to a total Chern number of zero. If the signs of the masses are uncorrelated, as would be expected for random disorder, then

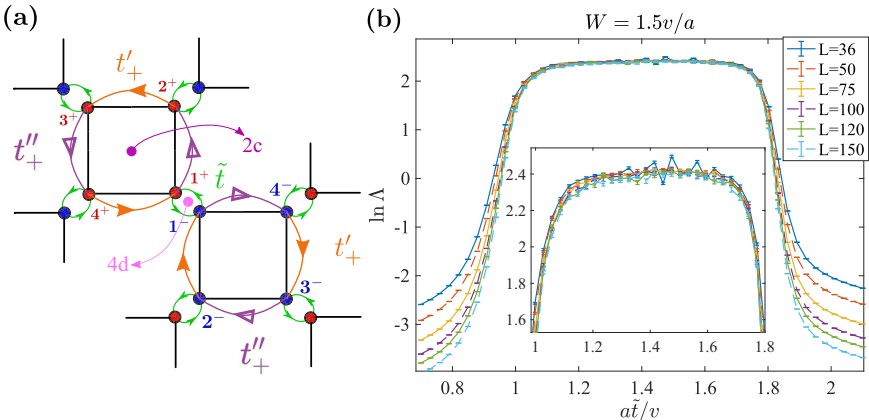

**Fig. 5 | The lattice model $H''_{8B}$. a** The unit cell and hoppings of $H''_{8B}$. **b** Normalized localization length $\Lambda$ as functions of $\tilde{t}$ with $A = 1.2, q = 0.625\exp(-0.15\pi i), W = 1.5v/a, E_F = 0$ and various transversal sizes L. The used longitudinal size is $M = 10^7$ and the precision ($\sigma_\Lambda/\Lambda$) has reached 1.5%.

$p_0 = \frac{(2N)!}{2^{2N}(N!)^2} \le \frac{1}{2}$, with the equality occurring only for $N = 1$. In our case $N = 2$ and $p_0 = \frac{3}{8}$. Thus, for weak disorder, we expect a percolating network of edge states. If the masses are positively correlated, $p_0$ could be further suppressed. To be concrete, $p_0 < \frac{1}{2}$ even in the $N = 1$ model if there is a higher possibility for the two masses to have the same sign. Under strong disorder, all local Chern insulators become trivial, and the system becomes localized.

The above argument can be immediately generalized to generic OAI transitions beyond $C_{2z}T$ symmetry. As a direct verification, we break $C_{2z}T$ in $H'_{8B}$ while keeping $m_{xy}$ and $C_{4z}T$ for the percolation mechanism and making two OAI limits at $a\tilde{t}/v = 0$ and $\infty$ inequivalent. The unit cell of the modified model $H''_{8B}$ is illustrated by Fig. 5a, where the orange arrows indicate hopping $t'_+ = (1+Ai)\frac{\pi v}{4a}$ and the purple arrows indicate $t''_+ = qt'_+ (q \in \mathbb{C})$. The magnetic space group of $H''_{8B}$ reduces to $P4'm'm$ (#99.165 in BNS setting), and the Wannier centers of OAI limits at $\tilde{t} = 0$ and $\infty$ are now located at the $C_{2z}$ centers 2c[2m′m′] and mirror planes 4d[m], respectively. The band structure also goes through an evolution of four Dirac points when $\tilde{t} = 0 \to \infty$. Figure 5b shows the localization length of $H''_{8B}$ with $A = 1.2, q = 0.625e^{-0.15\pi i}$. We can see that CMP indeed survives at $H''_{8B}$.

To further show the generality, in Fig. 6, we enumerate all the minimal OAI transitions protected by magnetic point groups $Pm, P2', P4'$, and $P6'$. The details of enumeration are summarized in Section II.B of Supplementary Information. Here, for a given symmetry group, the minimal OAI transition is defined as the OAI transition with minimal band deformation and molecular orbital transition. The band deformation of a generic OAI transition can be viewed as a superposition of minimal band deformations that can be band inversions at high symmetry points or gap closures at generic k-points (see the third column of Fig. 6). Also, since OAI can be wannierized to molecular orbitals (each orbital forming a site symmetry irrep at some Wyckoff position), an OAI transition can be characterized by occupation changes of these orbitals. The minimal molecular orbital transition refers to the minimal occupation change that can realize the corresponding band deformation. For example, the third row of block "P4′-NSOC" in Fig. 6 demonstrates a minimal OAI transition that replaces some occupied band forming irrep $X_2$ at $X$ point by a band with $X_1$ and moves an electron from irrep A at the Wyckoff position 1b to irrep A at 1a.

In terms of band deformation, these transitions can be divided into three categories: quadratic touching from a 2-dim irrep, Dirac points braiding, and immediate band gap closure-reopening. In the former two categories, the band structure is gapless for a finite parameter region in contrast to a single point in the last category. For the first category, if we add slow-varying disorders that mainly open a local

gap with a nontrivial Chern number ($p_0 < 1/2$), a CMP is highly possible due to the percolation mechanism protected by the average symmetry. For the second category, as discussed above, the possibility of CMP is higher with more Dirac points and stronger positive Dirac mass correlations. For the third category, the transition can go through a critical point at most since the gapless band structure is necessary for delocalization. The number and correlations of Dirac points also influence the possibility of delocalizing the gapless point.

## Discussion

For the first time, our work points out that transitions between trivial states (such as OAIs) without TRS can be critical under simple chemical potential disorders and, through a quantitative mapping, reveals that the criticality is due to a tricolor percolation mechanism of $C = 0, \pm 1$ regions. Since the chemical potential disorder is realistic and there are many topologically trivial magnetic materials with symmetries that forbid net (anomalous) Hall conductance[80,85], our work also has experimental relevance.

We notice that CMPs in 2D class A systems have been observed in previous works[57,86–88]. Ref. 86 did not report a CMP, yet we find its Fig. 2a may suggest a CMP similar to the one found by ref. 87. Ref. 87 reported a CMP in the Kane–Mele model ($\mathbb{Z}_2$ TI) in the presence of a weak Zeeman field and ascribed the criticality to the change of spin Chern number. Ref. 88 reported a CMP in the Bernevig–Hughes–Zhang model ($\mathbb{Z}_2$ TI) in the presence of a random magnetic field and ascribed the criticality to two coupled quantum Hall transitions. Therefore, these CMPs exist in topological phase transitions with additional weak perturbation terms. However, our CMP exists between two topologically trivial OAIs far from any topological state. The only difference between the two OAIs is the center and representation of Wannier states. It is also worth mentioning that, to exclude possible finite-size effects, we have verified the criticality of CMP up to system sizes $L = 2000$ for the network model and $L = 500$ for the lattice model (see Section I of Supplementary Information), which are larger than the system sizes $L = 24, 32, 128$ used in refs. 86–88, respectively.

## Methods

### Topological quantum chemistry of the $C_{2z}T$ model

The CMPs in the $C_{2z}T$ models arise between inequivalent OAIs. To depict the OAI transitions in the clean limit, we can use the tool of topological quantum chemistry, i.e., analyzing the transition of occupied magnetic element band representations (MEBRs)[80]. In our models, all the OAIs can be wannierized to molecular orbitals centering at the $C_{4z}$ and $C_{2z}T$ centers (Wyckoff positions 2b and 4c). These orbitals respect the site symmetries ($4m'm'$ and $2'm'm$ for 2b and 4c,

| (Magnetic) Space groups & Wyckoff postions | Minimal OAI transitions | Minimal band structure transitions | Minimal Wannier center transitions |
|---|---|---|---|
| Pm-NSOC  $1a\,[m]\,(x,0)$  $1b\,[m]\,\left(x,\tfrac{1}{2}\right)$ | $X_2 \to X_1$  $\begin{cases}\Delta^a_{A'} = 1 \text{ or } \Delta^a_{A''} = -1 \\ \Delta^b_{A'} = 1 \text{ or } \Delta^b_{A''} = -1\end{cases}$ | | $\pm 1$ $\pm 1$ |
| | $M_2 \to M_1$  $\begin{cases}\Delta^a_{A'} = 1 \text{ or } \Delta^a_{A''} = -1 \\ \Delta^b_{A'} = -1 \text{ or } \Delta^b_{A''} = 1\end{cases}$ | | $\pm 1$ $\pm 1$ |
| P2'-NSOC  $1a\,[2']\,(0,0)$  $1b\,[2']\,(1/2,0)$  $1c\,[2']\,(0,1/2)$  $1d\,[2']\,(1/2,1/2)$ | NULL  $\begin{cases}\Delta^{a/b/c/d}_A = 0, \pm 1 \\ \sum_{f=a}^{d}|\Delta^f_A| \neq 0\end{cases}$ | | $0,\pm1$  $0,\pm1$  $0,\pm1$  $0,\pm1$ |
| P4'-NSOC  $1a\,[4']\,(0,0)$  $1b\,[4']\,\left(\tfrac{1}{2},\tfrac{1}{2}\right)$  $2c\,[2]\,\left(\tfrac{1}{2},0\right),\left(0,\tfrac{1}{2}\right)$ | $\Gamma_1 \oplus \Gamma_1 \to \Gamma_2\Gamma_2$  $\begin{cases}\Delta^c_B = -\Delta^a_A \\ \quad = -\Delta^b_A = 1\end{cases}$ | | $+1$ $-1$  $-1$ $+1$ |
| | $M_1 \oplus M_1 \to M_2 M_2$  $\begin{cases}\Delta^c_A = -\Delta^a_A \\ \quad = -\Delta^b_A = 1\end{cases}$ | | $+1$ $-1$  $-1$ $+1$ |
| | $X_2 \to X_1$  $\{\Delta^a_A = -\Delta^b_A = 1$ | | $0$ $-1$  $+1$ $0$ |
| P4'-SOC  $1a\,[4']\,(0,0)$  $1b\,[4']\,\left(\tfrac{1}{2},\tfrac{1}{2}\right)$  $2c\,[2]\,\left(\tfrac{1}{2},0\right),\left(0,\tfrac{1}{2}\right)$ | $X_3 \to X_4$  $\begin{cases}\Delta^a_{\overline{1}\overline{E}^2\overline{E}} = \pm 1 \text{ or } \Delta^b_{\overline{1}\overline{E}^2\overline{E}} = \pm 1 \\ \Delta^c_{\overline{1}\overline{E}} = -1 \text{ or } \Delta^c_{\overline{2}\overline{E}} = 1\end{cases}$ | | $\pm1$ $0,\pm2$  $0,\pm2$ $\pm1$ |
| | NULL  $\{\Delta^a_{\overline{1}\overline{E}^2\overline{E}} = -\Delta^b_{\overline{1}\overline{E}^2\overline{E}} = 1$ | | $0$ $\mp2$  $\pm2$ $0$ |
| P6'-NSOC  $1a\,[6']\,(0,0)$  $2b\,[3]\,\left(\tfrac{1}{3},\tfrac{2}{3}\right),\left(\tfrac{2}{3},\tfrac{1}{3}\right)$  $3c\,[2']\,\left(\tfrac{1}{2},0\right),\left(0,\tfrac{1}{2}\right),$  $\left(\tfrac{1}{2},\tfrac{1}{2}\right)$ | $\Gamma_1 \oplus \Gamma_1 \to \Gamma_2\Gamma_3$  $\{\Delta^a_{1E\,^2E} = -\Delta^b_{A_1} = 1$ | | $-1$ $-1$  $0$ $0$ $0$  $+2$ |
| | $K_1 \oplus K_1 \to K_2 K_3$  $\{\Delta^a_{1E\,^2E} = -\Delta^b_{2E} = 1$ | | $-1$ $-1$  $0$ $0$ $0$  $+2$ |
| | $K'_1 \oplus K'_1 \to K'_2 K'_3$  $\{\Delta^a_{1E\,^2E} = -\Delta^b_{1E} = 1$ | | $-1$ $-1$  $0$ $0$ $0$  $+2$ |
| | NULL  $\begin{cases}\Delta^a_{A_1} = \Delta^a_{1E\,^2E} \\ \quad = -\Delta^c_A = \pm 1\end{cases}$ | | $0$ $0$  $\pm1$ $\pm1$ $\pm1$  $\mp3$ |

respectively) and hence can be characterized by the irreps of site symmetries. According to the topological quantum chemistry, these molecular orbitals will induce bands with MEBRs listed in Table 2. One MEBR is the minimal group of bands formed by a type of molecular orbitals, and the left part of an MEBR notation indicates the site symmetry irrep of the orbitals, e.g., $A_b{\uparrow}G$ indicates the MEBR formed by $s$ orbitals (trivial irrep $A$) at positions $2b$. Therefore, we can deduce the

Wannier centers of OAIs from the occupied MEBRs, depicting the OAI transitions by MEBR transitions.

We start with the network model. Although a network model has infinitely many occupied bands, these bands form an infinite direct sum of MEBRs and can be viewed as a special kind of OAI. The band structure comprises disconnected branches, each containing two bands. One branch forms one of the following four MEBRs defined in

**Fig. 6 | OAI transitions in (magnetic) space groups _Pm_,_P2′_,_P4′_,_P6′_.** The first column contains symbols of the (magnetic) space groups, appearance of SOC, and the Wyckoff positions except for the general positions. The "-NSOC" indicates the absence of SOC in contrast to "-SOC". For symmetry groups having the same transition features with and without SOC, we only list the case without SOC. The square brackets [·] and parentheses (·) contain the site symmetries and coordinates of the Wyckoff positions, respectively. The second column contains the reciprocal and real space information of the minimal OAI transitions defined in subsection "OAI transitions in generic magnetic point groups". The transition with a symbol of momentum irrep exchange $R_n \rightarrow R_m$ induces band inversion(s) at high symmetry point $R$ (and its inequivalent symmetric partners), which replaces some occupied band(s) forming $R_n$ by band(s) with $R_m$. The transition with "NULL" closes the band

gap at general k-points. Equations inside "{" depict the minimal orbital occupation changes, where $\Delta_Q^f$ denotes the occupation change of irrep Q at each Wyckoff position $f$. The particle conservation is not explicitly stated, yet it is enforced in every transition by default. The third column illustrates the transitions in reciprocal space. A plot with a Brillouin zone is decorated by red crosses and arrows that indicate the positions and movements of Dirac points, respectively. A plot with four curves suggests that the band inversion will go through a gapless region where a quadratic touching from a 2-dim irrep dominates the physics near the Fermi level. The last column illustrates the possible minimal changes of Wannier centers during the transitions. The colored dots indicate the Wyckoff positions inside one cell, and the colored numbers show the possible occupation changes of each position.

---

**Table 2 | MEBRs of $P_C4bm$ (#100.177 in BNS setting) involved in this work, taken from the MBANDREP program on the Bilbao Crystallographic Server[80]**

| Wyckoff pos. | 2b $(\tfrac{3}{4}\tfrac{1}{4}0),(\tfrac{1}{4}\tfrac{3}{4}0)$ | | | | 4c $(000),(0\tfrac{1}{2}0),(\tfrac{1}{2}00),(\tfrac{1}{2}\tfrac{1}{2}0)$ |
|---|---|---|---|---|---|
| Site sym. | $4m'm',4$ | | | | $2'm'm,m$ |
| MEBR | $A_b{\uparrow}G$ | $B_b{\uparrow}G$ | $^1E_b{\uparrow}G$ | $^2E_b{\uparrow}G$ | $A''_c{\uparrow}G$ |
| Orbital | 1 | $d_{x^2-y^2}+id_{xy}$ | $p_x+ip_y$ | $p_x-ip_y$ | $p_y$ |
| Irreps at Γ | $\Gamma_1\oplus\Gamma_4$ | $\Gamma_2\oplus\Gamma_3$ | $\Gamma_5$ | $\Gamma_5$ | $\Gamma_2\oplus\Gamma_4\oplus\Gamma_5$ |
| Irreps at M | $M_5$ | $M_5$ | $M_2\oplus M_3$ | $M_1\oplus M_4$ | $M_2\oplus M_4\oplus M_5$ |
| Irreps at X | $X_1$ | $X_1$ | $X_1$ | $X_1$ | $2X_1$ |

The real space orbital character of each MEBR is shown in the "Orbital" row. For example, the MEBR $B_b{\uparrow}G$ can be generated by an $d_{x^2-y^2}+id_{xy}$ type orbital at the first 2b position $(\tfrac{3}{4},\tfrac{1}{4},0)$. Note that $A''_c{\uparrow}G$ is the only decomposable MEBR, although bands with this MEBR are always connected in this work. One should notice that we use a different convention of the origin point as the Bilbao Crystallographic Server.

---

Table 2: $A_b{\uparrow}G$, $B_b{\uparrow}G$, $^1E_b{\uparrow}G$, $^2E_b{\uparrow}G$, result from effective $s,d_{x^2-y^2}+id_{xy}$ (or equivalently $d_{x^2-y^2}-id_{xy}$), $p_x-ip_y, p_x+ip_y$ orbitals, respectively. All these orbitals center at the Wyckoff position 2b. Since 2b has multiplicity 2 (two $C_{4z}$ centers per unit cell), each MEBR contains two bands. Also, notice that the band structure comprises infinite repeating units. Each unit contains eight bands, and one can generate the whole band structure by energy translations of a unit with step $2\pi v/a$. Hence, we can focus on the upper eight of the ten bands in Fig. 1g–i. When $\theta < 0$, direct gaps separate different branches, and the lower four of the focused eight bands form a direct sum of two MEBRs: $^2E_b{\uparrow}G \oplus A_b{\uparrow}G$. When $\theta = 0$, the gaps close, and the band structure is indeed that of a ballistic 1D metal with linear dispersion relations. As $\theta$ increases across 0, irreps $\Gamma_1, M_1$ ($\Gamma_2, M_2$) go up (down) across the energy level. Similar irrep exchanges also happen above and below the repeating unit, e.g., the lower two of the focused eight bands will exchange irreps with the lower bands that exceed the scope of Fig. 1g–i. After the transition ($\theta > 0$), the gaps reopen, and the MEBRs of the lower four of the focused eight bands change to $^1E_b{\uparrow}G \oplus B_b{\uparrow}G$. Hence, $\theta = \pm\pi/2$ correspond to two inequivalent trivial phases, although the molecular orbitals of both phases center at the $C_{4z}T$ centers.

We now turn to the band evolution of $H_{8B}$ ($H'_{8B}$ is similar). See Fig. 2c–e, as $\tilde{t}$ increases from 0 to $\pi v/a$, $\Gamma_1$ and $M_1$ rise across the Fermi level while $\Gamma_2$ and $M_2$ fall below the Fermi level. We encounter the same irreps exchange to the network model. Despite similarities near the Fermi level, the MEBR transition of $H_{8B}$ is different from the network model. The lower four bands of $H_{8B}$ change from $^2E_b{\uparrow}G \oplus A_b{\uparrow}G$ to $A''_{c\uparrow}G$ during the transition. The Wyckoff position of $A''_{c\uparrow}G$ is 4c, which is the $C_{2z}T$ center rather than the $C_{4z}$ center (Table 2). This difference in molecular orbital transition is unavoidable since the transition in the network model involves exchanges of representations between different repeating units, while $H_{8B}$ only has one unit. In the network model with a sufficient number of bands, the four bands below the Fermi level not only exchange representations with the bands above them but also with the bands (in another repeating unit) below them. However, $H_{8B}$ has no band below the lower four bands. Nevertheless, since the origin of this difference is well below the Fermi level, it should

not affect the low-energy physics. Therefore, we can expect similar low-energy behaviors between $H_{8B}$ and the network model, which our numerical data confirms.

It is worth mentioning that the transition in $H_{8B}$ changes the position of the MEBRs from $C_{4z}$-centers (2b) to $C_{2z}T$-centers (4c). No $C_{2z}T$ center is occupied before the transition, and the RSI $\delta_w = 0$. Given that there are four $C_{2z}T$ centers per cell and four occupied bands, every $C_{2z}T$ center is occupied by one electron after the transition, and the system has RSI $\delta_w = 1$. Therefore, the second Stiefel-Whitney class $w_2$[60,66] must change from 0 to 1. As we have discussed in the second paragraph of subsection "A $C_{2z}T$-symmetric quantum network model", the transition process must involve braiding of the Dirac points. In addition, although the lower four bands form $A''_{c\uparrow}G$ are always connected in our models, in general cases, $A''_{c\uparrow}G$ can be decomposed into two fragile topological bands ($\Gamma_5, M_5, X_1$) and two trivial bands (forming MEBR $A_2{\uparrow}G$ with Wyckoff position 2a $(\tfrac{1}{4}\tfrac{1}{4},0),(\tfrac{3}{4}\tfrac{3}{4},0)$, i.e., the centers of white squares in Fig. 1c), which is expected from $w_2 = 1$.

## Quasi-1D localization length and transfer matrix method

A commonly used physical quantity in research of localization is the quasi-1D localization length $\rho_{q-1D}$. It is defined on a 2D/3D sample prepared in a quasi-1D shape, e.g., a long, thin cylinder with $L_{axial} \gg L_{radius}$. $\rho_{q-1D}$ reflects the decaying rate of eigenstates in the quasi-1D direction, e.g., the axial direction of a long thin cylinder. Since any 1D system is localized under nonzero disorder strength, $\rho_{q-1D}$ will always be finite except for a perfectly clean sample. Localization of the original 2D/3D system (in a 2D/3D shape) can be derived from a scaling analysis of the dimensionless quasi-1D localization length $\Lambda = \rho_{q-1D}/L$, where $L$ is the transversal size of the quasi-1D sample. We denote the localization length of a normally shaped (scales of different directions are similar) and sufficiently large sample as $\rho$. For a metallic system, $\rho$ in a (normally shaped and sufficiently large) sample is much larger than the sample size. Thus, $\rho_{q-1D}(L)$ increases faster than $L$, i.e., $\Lambda(L) \rightarrow \infty$ in the limit $L \rightarrow \infty$. For an insulating system, $\rho$ is finite in a (normally shaped and sufficiently large) sample. Hence, $\rho_{q-1D}$ will converge to $\rho$ when $L \gg \rho$, i.e., $\Lambda(L) \rightarrow 0$ as $L \rightarrow \infty$. In practice,

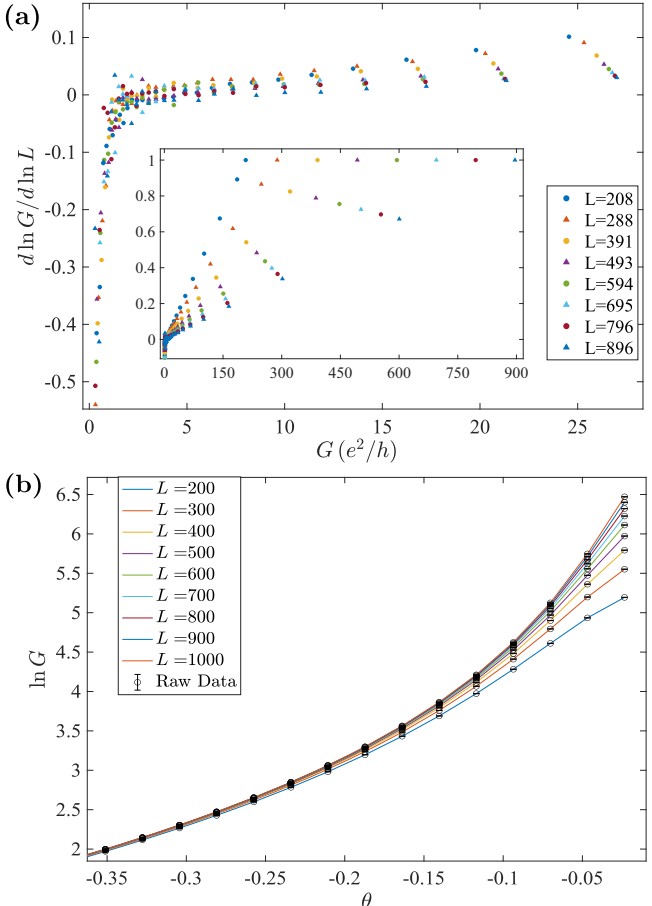

**Fig. 7 | $\beta$-function and conductance fitting near the ballistic limit. a** The $\beta$-function of the network model derived from finite differences of conductance data in Fig. 1f. The legends inform the central sizes of the finite differences. The insect is the zoom out of the $\beta$-function for large $G$. **b** Conductance fitting according to the linear hypothesis of $\beta$-function for large $G$. Colored curves are fitting results while the circles are raw data. The goodness of fit has reached 0.23 and will become higher as getting closer to the ballistic limit. (1 corresponds to a perfect fitting and 0.05 is the frequently used threshold of accepting a fitting hypothesis).

we identify the region where $\Lambda(L)$ monotonically increases as the metallic phase and where $\Lambda(L)$ monotonically decreases as the localized phase. If a system contains both localized and extended phases in some parameter space, there will be a critical region (usually a boundary with measure zero) in the parameter space where $\Lambda(L)$ is independent of (sufficiently large) $L$.

The transfer matrix method[77] is a widely used numerical approach in calculating $\rho_{q-1D}$. Although it has different formulae for (generic) network and lattice models, the basic ideas are the same. A quasi-1D sample is divided into layers with normals along the quasi-1D direction. The amplitudes of an energy-eigenstate on different layers are related by the Schrodinger equation. A ($2s \times 2s$ shaped) transfer matrix $T_n$ generally transforms the amplitudes on the $(l - r_1)$th - $(l + r_2 - 1)$th layers to those on the $(l - r_1 + 1)$th - $(l + r_2)$th layers. Here, $s \in \mathbb{N}^+$ is proportional to the number of degrees of freedom in one layer. And $r_1, r_2 \in \mathbb{N}^+$ represent that only the $(l - r_1)$th - $(l + r_2)$th layers can hop/propagate (in one step) to the $l$ th layer. The values of $r_1$ and $r_2$ depend on the hopping of concrete models. Since the transfer matrix of a (general) network model is determined by the transmission matrix, the amplitudes in the $(l+1)$th layer depend only on the $l$th layer. Hence, $r_1 = r_2 = 1$ for general network models (not only ours). For general lattice models, $r_1$ and $r_2$ can take arbitrary finite non-negative integer values. Nevertheless, in our models, $r_1 = r_2 = 1$.

To extract $\rho_1 D$ from the transfer matrix, we can consider a consecutive product of transfer matrices $O_M = \prod_{n=1}^{M} T_n$. Because of the disorder, some elements in $T_n$ are random variables. According to the Oseledec theorem, the limit $P = \lim_{M \to \infty} (O_M^\dagger O_M)^{1/2M}$ exists and has eigenvalues $\{\exp(v_1), \exp(-v_1), ... \exp(v_s), \exp(-v_s)\}$ where $v_i \geq v_{i+1} \geq 0$, $i = 1, 2 ... s$. These (positive) exponents are so-called Lyapunov exponents (LEs). The definition of $P$ indicates that an eigenvector $\boldsymbol{\eta_i}$ of $P$ with eigenvalue $\exp(-v_i)$ satisfies $\| O_M \boldsymbol{\eta_i} \|^2 = \boldsymbol{\eta_i}^\dagger (O_M^\dagger O_M) \boldsymbol{\eta_i} = \boldsymbol{\eta_i}^\dagger [(O_M^\dagger O_M)^{1/2M}]^{2M} \boldsymbol{\eta_i} \approx \boldsymbol{\eta_i}^\dagger P^{2M} \boldsymbol{\eta_i} = \| \exp(-M v_i) \boldsymbol{\eta_i} \|^2$ for sufficiently large M. Therefore, the smallest LE $v_s$ determines the decaying rate of energy-eigenstates (along the quasi-1Dc direction) since any energy-eigenstate is a superposition of eigenvectors of $P$ with eigenvalues $\exp(-v_i)$ (the amplitudes cannot grow exponentially hence $\exp(v_i)$ is excluded). Because of this, one can define the quasi-1D localization length as the inverse of the smallest LE: $\rho_{q-1D} = 1/v_s$.

Due to the space limitation, there are three main aspects we cannot explain here: calculating conductance from the transfer matrix, the technique for numerical stability of LE, and the concrete formulae of the transfer matrices of our network and lattice models. Readers interested in these details can refer to Section V of Supplementary Information.

### $\beta$-function of the CMP

The $\beta$-function $\beta = d \ln G / d \ln L$ is an additional quantity to verify the criticality of the observed delocalized phases. Figure 7a shows the $\beta$-function of the network model derived from the finite difference method (we have ignored the default conductance unit $e^2/h$). As we can see, all the data points fall into two parts corresponding to the delocalized and localized phases. A critical conductance $G_c = 2$-3 divides these two phases. In the localized phase below $G_c$, data points collapse to one curve, demonstrating the Anderson localization. In the delocalized phases above $G_c$, the data points distribute around zero below $G \sim 5$. And above $G \sim 5$, data from different sizes deviate from each other and become significantly positive. These positive data points result from the finite size effect of the ballistic limit at $\theta = 0$. To justify that the delocalized phase with $-\pi/4 \lesssim \theta < 0$ is indeed a CMP, we have to prove that $\beta(G > G_c, L \to \infty) \to 0^+$.

When $\theta = 0$, the network degenerates to decoupled parallel chiral wires in four directions ($\pm \hat{x}, \pm \hat{y}$). Since there are $L$ channels in one direction, the ballistic conductance $G_{max}(L) = L$. Hence, for a given system size $L$, the $\beta$-function should terminate at the point ($G = L, \beta = 1$), which is confirmed numerically (the inset of Fig. 7a). Due to the finite size effects, the ballistic limit will induce a diffusive metal phase with $\theta \to 0$ corresponding to the positive region of $\beta$-functions for large $G$. Further, the data suggest a linear hypothesis of $\beta$-function for large $G$, i.e., $\beta(G, L) = G/L$ when $G \gg G_c$. To verify this hypothesis, we can view it as a differential equation and test its solution (as a hypothesis of conductance) on conductance data. To be concrete, $\beta(G, L) = G/L$ implies an ansatz of conductance $G(\theta, L)^{-1} = G(\theta, \infty)^{-1} + L^{-1}$ when $G \gg G_c$. Here, the fitting parameter $G(\theta, \infty)$ is the conductance in the thermodynamic limit for a given $\theta$. As illustrated in Fig. 7b, this ansatz fits our conductance data very well when $G > 5$. Although we have not understood the mechanism behind the linear $\beta$-function, we can conclude from the above results that $\beta(G > G_c, L \to \infty) \to 0^+$, proving the criticality of CMP.

### Localized OAI transition in class AI

One should not conclude from the above results that CMP exists in any transition between inequivalent 2D OAIs. Due to the localized nature of OAI, it is commonly believed that disorders should localize general transitions between inequivalent 2D OAIs. We can obtain a localized OAI transition by modifying our lattice model $H_{8B}$. Recall Eq. (4), $H_{8B}$ is described by four real parameters $t, t', \tilde{t}$, and $\phi_{pq}$. The only parameter breaking TRS is $|\phi_{pq}| = 3\pi/4$. If we take $|\phi_{pq}| = 0$ or $\pi$, the system will

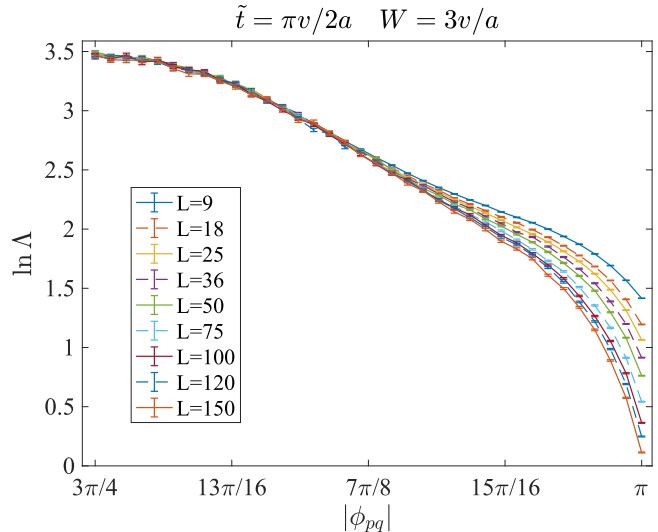

$$\tilde{t} = \pi v/2a \quad W = 3v/a$$

**Fig. 8 | Normalized localization length Λ as functions of $|\phi_{pq}|$ with**
$t = \frac{\pi v}{2a}, W = 3v/a, E_F = 0$, **and various transversal sizes $L$.** The used longitudinal size
is $M = 10^7$ and the precision $(\sigma_N/\Lambda)$ has reached 2%. The point with $|\phi_{pq}| = 3\pi/4$
corresponds to the middle of Fig. 2f. The point with $|\phi_{pq}| = \pi$ recovers TRS and is
localized under any finite on-site disorders.

respect TRS even if on-site disorders are present. Now, the previous
$C_{2z}T$ centers are promoted to $C_{2z}$ centers. According to ref. 64, in the
presence of TRS, the Wyckoff position with site symmetry 2 has a
$\mathbb{Z}$-valued RSI $\delta = m_+ - m_-$ where $m_+$ $(m_-)$ is the occupation number of
orbital even (odd) under $C_{2z}$. Hence, when $|\phi_{pq}| = 0$ or $\pi$, tuning $\tilde{t}$ from
$0$ $(\delta = 0)$ to $+\infty$ $(\delta = -1)$ still drives a transition between inequivalent
OAIs. On the other hand, due to the weak localization effect in the
presence of spinless TRS, the system must be localized regardless of
$t, t'$, and $\tilde{t}$. In Fig. 8, we take $\tilde{t} = \frac{\pi v}{2a}$, corresponding to the middle of the
CMP in Fig. 2f, and tune $|\phi_{pq}|$ from $3\pi/4$ to $\pi$. We can see a CMP-LP
transition in Fig. 8. For all the $|\phi_{pq}|$ scanned, the band structure in the
clean limit is gapless near the Fermi level $E_F = 0$, i.e., the clean system
stays at the metallic intermediate state of the OAI transition but will be
localized by disorders. Although the LP shrinks with weaker disorder
strength $W$, $|\phi_{pq}| = \pi$ is always localized.

## Data availability
We provide all the raw data in Supplementary Dataset. Source data are
provided with this paper.

## Code availability
Codes required for reproducibility are available upon request to the
authors.

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

## Acknowledgements

We are grateful to Roni Ilan, Ryuichi Shindou, Chen Wang, Chui-Zhen Chen, Jian Li, and Anton Akhmerov for helpful discussions. Z.-D.S. and F.-J.W. were supported by National Natural Science Foundation of China (General Program No. 12274005), Innovation Program for Quantum Science and Technology (No. 2021ZD0302403), National Key Research and Development Program of China (No. 2021YFA1401903). B.A.B. was supported by the European Research Council (ERC) under the European Union's Horizon 2020 research and innovation program (grant agreement No. 101020833), the ONR Grant No. N00014-20-1-2303, the Schmidt Fund for Innovative Research, Simons Investigator Grant No. 404513, the Gordon and Betty Moore Foundation through the EPiQS Initiative, Grant GBMF11070 and Grant No. GBMF8685 towards the Princeton theory program. Further support was provided by the NSF-MRSEC Grant No. DMR-2011750, BSF Israel US foundation Grant No. 2018226, the Princeton Global Network Funds. R.Q. is supported by the Simons Foundation award 990414, the U.S. Department of Energy (DOE) under award DE-SC0019443, and the NSF MRSEC DMR-2011738

## Author contributions

Zhi-Da Song, Ady Stern, and B. Andrei Bernevig are conceived of this work. Zhi-Da Song proposed the network model and the percolation argument. Fa-Jie Wang completed the mapping from the network model to the lattice model and ran the numerical calculations. Zhen-Yu Xiao proposed the calculation on local Chern marker. Raquel Queiroz helped analyze the β-function. Ady Stern proposed the connection between the number of Dirac points and the percolative conductivity. All the authors contributed to the writing.

## Competing interests

The authors declare no competing interests.
