## [Peer Review File · Nature Communications]

Anderson Critical Metal Phase in Trivial States Protected by Average Magnetic Crystalline SymmetryREVIEWER COMMENTS

Reviewer #1 (Remarks to the Author):

This paper deals with gapless phase that appears at the phase transition region between the Stiefel-Whitney nontrivial (C2zT-symmetry-protected obstructed atomic} insulator and the trivial one. The main finding of this work is that that this gapless becomes a critical metallic phase (CMP) under the random chemical potential with average C2zT symmetry. Furthermore, the authors show that the quantum-critical description of the CMP can be mapped into a percolation network model, with the regions featuring the trivial or unit absolute value Chern number, and the C2zT symmetry preserved on average. To numerically study this problem, the authors first consider a quantum network model on the Manhattan lattice. Subsequently, this model is regularized to an eight-band lattice model (to be able to explicitly feature the band topology) finding such a C2zT protected CMP and confirm the qualitative critical behavior as predicted from the previous network model.

The problem studied in the manuscript is of a very specialized nature, studying the behavior with the disorder of a phase with a very limited both theoretical and experimental relevance. Furthermore, the model studied here, as also authors explicitly mention in the Discussion section, shares the same universality class as random Kane-Mele and Bernevig-Hughes-Zhang model, since both of them are with C2zT symmetry but breaking time-reversal, which is the hallmark feature of the phase studied here (hence the title of the manuscript). This is expected from the universality hypothesis (symmetries, dimensionality and range of the disorder and/or interaction determine the universality class). Criticality of the disordered C2zT KM and BHZ models has been found in Refs. 74, 75 and 48, 76, respectively, as also claimed by the authors in the Discussion section.

Given the above, I find that this manuscript does not cut a high bar imposed by Nature Communications. It is more appropriate to a specialized journal, such as Scientific Reports in Nature family, or Physical Review B. In my opinion, the readers would benefit tremendously if the supplementary materials have been organized more concisely. Furthermore, I have the following technical points. To completely characterize the disorder at the transition between the CMP and localized phase, the authors should consider the following:

1. Calculate the critical exponents for the correlation length and the conductivity at this transition.
2. At least discuss qualitatively, the appropriate continuum quantum field theory describing the transition from the CMP to the localized phase. In particular, find the critical exponents in this theory and compare with the ones obtained numerically.

REVIEWER COMMENTS

Reviewer #2 (Remarks to the Author):

In this work, the authors propose and study a collection of related models in which an intervening critical metal phase is found as a C2T-protected obstructed atomic insulator is tuned into another localized insulator by disorder which respects the C2T symmetry on average. The analysis began with a well-motivated network model on the Manhattan lattice, which is then regularized and truncated into two related lattice models. The realization of an OAI phase is verified from symmetry analysis. The presence of a critical metal phase as disorder is introduced is argued on the grounds of a percolation picture in terms of the network model, and is also evidenced by numerical results on both the network and lattice models.

Whether an intervening critical metal phase could exist in the disorder-induced transition between topologically trivial phases is an interesting problem, and I am convinced that the authors have provided a positive example involving an OAI on one side. However, there is one question I believe the authors should address before I could make any recommendations: from the authors' general discussion, it was suggested that the intervening critical metal phase could be expected as a general phenomenon as the C2T-protected OAI is disorder-tuned into another localized phase. Yet, the models studied by the authors appeared to be more tailor-made than what would support the general claim. As the authors discussed the behavior of the network model could be understood from the percolation problem of the random flux model. Since the lattice models descended from the network model, it is unclear if they are sufficiently general to reveal the generic behavior of how the C2T-protected OAI responds to on-site potential disorder. For instance, while it is neat that the random vector potential disorder in the network model could be mapped to potential disorder (by dropping the smaller off-diagonal terms and the sub-leading hopping disorder), the potential disorder one obtains at the end appears to be locally correlated. More specifically, from Eqs. (B41)-(B45) in the supplementary information, it appears that the vector potential disorder A , which are independent on different edges, ultimately enters into the on-site disorder term of the various corner orbitals in a unit cell through a very specific combination.

If the authors' claim is indeed general, then any on-site disorder which respects C2T on average which tunes the OAI into the targeted localized phase should similarly induce the intervening critical metal phase. Is this true? Alternatively, the authors may explain why the perceived local correlation in the potential disorder obtained is absent and there is no correlation beyond requiring the disorder ensemble to respect C2T on average.

Lastly, a minor comment: in writing $E_n(k) = E_{n+8}(k)$ above Eq. (3) of the main text, it may be clearer to mention the shift in energy/ repeating window.

Reviewer #3 (Remarks to the Author):

The past decade has seen significant advancements in understanding of topological phases protected by crystalline symmetries. Recently, intriguing characteristics of topological crystalline phases have been discovered. Along this line, the authors presented evidence for a critical metallic

phase between an obstructed atomic insulator and a genuine trivial one.

Although I found the presented manuscript interesting, I have concerns about the novelty of the findings presented in the manuscript. More specifically, I wonder if experts in the field might easily anticipate the critical metallic phase in this work. Furthermore, I am concerned that the presented results are not as impactful as the journal expects. This expectation is based on the following three things:

(i) As the authors mentioned in Introduction, Ref. 15 has established a connection between localization/delocalization and the classification of massive/massless Dirac Hamiltonians (equivalently, K-theory classification). Also, Ref. 15 has extended the discussion to the case with reflection symmetry. Furthermore, the classification of Dirac Hamiltonians in the presence of point group symmetries has been established in arxiv:1811.01977 and 1907.09354. Importantly, K-groups often contain atomic insulators whose electrons are localized at some points in real space.

(ii) According to recent classification results, the K-group is $(\mathbb{Z}_2)^3 \times \mathbb{Z}$ in magnetic layer group $p112'$, where all generators possess localized electrons. In this symmetry setting, there are four inequivalent Wyckoff positions: $(x,y) = (0,0)$, $(1/2, 0)$, $(0, 1/2)$, and $(1/2, 1/2)$. In fact, roughly speaking, generators of $(\mathbb{Z}_2)^3$ correspond to the "difference" between two insulators with localized electrons at two different Wyckoff positions. In this case, we can say that nontrivial elements of $(\mathbb{Z}_2)^3$ are obstructed atomic insulators with some appropriate choice of a reference state in K-theory.

(iii) In Ref. 39, some of the authors have revealed that such a metallic phase exists between a trivial insulator and a nontrivial element of the K-group (an axion insulator) when inversion symmetry is respected on average.

The first two points imply that the emergence of such critical metallic phases is a common feature of phase transitions between two different elements of a K-group. Furthermore, the third point has already pointed out the localization/delocalization protected by crystalline symmetries on average. I respectfully ask the authors to clarify whether the critical metric phase in this work is completely different from those among distinct elements of K-groups.

On the side of the presentation, I would like to ask the authors to consider the following comments/questions.

I could not find the definition of "respecting symmetries on average." Since this is a key concept of this work, the authors should add a precise definition somewhere.

Considering the current length of the manuscript and the existence of Method section, I recommend them to move some technical details from Supplementary Information to Method or main text. For example, in Figures 1 and 2, the authors use symbols of irreducible representations. I think, although the manuscript is well written, the manuscript would be more readable when Tables I and II are included in Method.

List of revisions made

1. To satisfy the convention of Nature Communications, we divide the main text into four sections: Introduction (I), Results (II), Discussion (III), and Methods (IV).
2. In reply to the Referees' common concern about the general significance of our results, we add new content about CMPs beyond systems respecting $C_{2z}T$ symmetry. In Sec.II.D, we briefly discuss the OAI transitions and CMPs in generic (magnetic) point groups and enumerate the cases in $Pm, P2, P4'$, and $P6'$. We offer the technical details of enumeration in Appendix II.B. In Sec.II.D, we also break the average $C_{2z}T$ in the lattice model H'_{8B} and find that the CMP survives. In Appendix II.A, we show CMP in a Kagome network model. Considering the generality of CMP suggested by these new results, we revise the title from "Anderson Critical Metal Phase in Trivial States Protected by C2zT Symmetry on Average" to "Anderson Critical Metal Phase in Trivial States Protected by Average Magnetic Crystalline Symmetry." We also rephrase the Abstract, Introduction, and Discussion according to the generality of CMP. The existence of CMPs beyond $C_{2z}T$ is also mentioned in the discussion of the Manhattan network model (Sec.II.A).
3. In reply to Referee B and C's question about the universal appearances of CMP in OAI transitions respecting $C_{2z}T$, we emphasize the common belief of OAI transitions being localized and offer a localized example in Sec. IV.D.
4. In reply to Referee A's worry about the novelty of our CMP, we emphasize the fundamental differences between CMPs found in previous works and ours in Sec. III. We also emphasize in Sec. I that our CMP is unpredictable for conventional theories, including the universality hypothesis mentioned by Referee A.
5. In reply to Referee A's question about the nature of critical-localized transition, we investigate the beta-function of the Manhattan network model in Sec. IV.C. The result and possibility of a BKT-like transition are mentioned above Eq. (2).
6. According to Referee A's suggestion on the readability of supplementary materials, we reorganize the supplementary materials by moving the supplemental data before the technical details.
7. In reply to Referee B's question about disorder correlation being ignored, we explain in Sec. II.B that the correlation results in no qualitative difference and attach the data in Appendix IV.F.
8. We fix the typo of band energy periodicity pointed out by Referee B. The incorrect $E_n(k) = E_{n+8}(k)$ is replaced by $E_n(k) = E_{n+8}(k) + 2\pi v/a$.
9. As Referee C requires, we add a definition of the average symmetry in Sec. I.

10. According to the last two suggestions of Referee C, We copy Tables I & II in the supplementary materials to the main text and introduce the techniques of topological quantum chemistry and transfer matrix in Secs. IV.A & IV.B, respectively.
11. We delete the discussion about the fragile topology because we realized some subtle differences between OAI and fragile states. We hence leave it for future works.
12. We update Figs. 1(e) & (f) in the main text and Figs. 1 & 2 in the supplementary materials with new data in larger transversal sizes.
13. We make up the data precisions of Figs.1 & 2 in the main text, which are missing in the original manuscript.
14. We make minor improvements in the readability and consistency of terminologies.
15. We add the following references to the main text:

1. Eslam Khalaf, Hoi Chun Po, Ashvin Vishwanath, and Haruki Watanabe, "Symmetry indicators and anomalous surface states of topological crystalline insulators," [arXiv:1711.11589 \[condmat, physics:hep-th\] \(2017\)](https://arxiv.org/abs/1711.11589), [arXiv: 1711.11589](https://arxiv.org/abs/1711.11589).

2. Zhida Song, Tiantian Zhang, Zhong Fang, and Chen Fang, "Quantitative mappings between symmetry and topology in solids," [Nature Communications 9, 3530 \(2018\)](https://doi.org/10.1038/s41467-018-03530-2).

3. Zhida Song, Sheng-Jie Huang, Yang Qi, Chen Fang, and Michael Hermele, "Topological states from topological crystals," [Science Advances 5, eaax2007 \(2019\)](https://doi.org/10.1126/sciadv.aax2007).

4. Eyal Cornfeld and Adam Chapman, "Classification of crystalline topological insulators and superconductors with point group symmetries," [Physical Review B 99, 075105 \(2019\)](https://doi.org/10.1103/PhysRevB.99.075105), publisher: American Physical Society.

5. Ken Shiozaki, "The classification of surface states of topological insulators and superconductors with magnetic point group symmetry," [Progress of Theoretical and Experimental Physics 2022, 04A104 \(2022\)](https://doi.org/10.1063/1.514104).

6. Jiong-Hao Wang, Yan-Bin Yang, Ning Dai, and Yong Xu, "Structural-disorder-induced second-order topological insulators in three dimensions," [Phys. Rev. Lett. 126, 206404 \(2021\)](https://doi.org/10.1103/PhysRevLett.126.206404).

7. Yan-Bin Yang, Kai Li, L.-M. Duan, and Yong Xu, "Higher-order topological anderson insulators," [Phys. Rev. B 103, 085408 \(2021\)](https://doi.org/10.1103/PhysRevB.103.085408).

8. Chang-An Li, Song-Bo Zhang, Jan Carl Budich, and Björn Trauzettel, "Transition from metal to higher-order topological insulator driven by random flux," [Phys. Rev. B 106, L081410 \(2022\)](https://doi.org/10.1103/PhysRevB.106.L081410).

9. I. C. Fulga, B. van Heck, J. M. Edge, and A. R. Akhmerov, "Statistical topological insulators," [Phys. Rev. B 89, 155424 \(2014\)](https://doi.org/10.1103/PhysRevB.89.155424).

10. Ruochen Ma and Chong Wang, "Average symmetry protected topological phases," [Physical Review X 13 \(2023\), 10.1103/physrevx.13.031016](https://doi.org/10.1103/PhysRevX.13.031016).

11. "Supplementary materials" .

12. J M Kosterlitz and D J Thouless, "Ordering, metastability and phase transitions in two-dimensional systems," *Journal of Physics C: Solid State Physics* 6, 1181 (1973).

13. Shou-Cheng Zhang and Daniel P. Arovas, "Effective field theory of electron motion in the presence of random magnetic flux," *Physical Review Letters* 72, 1886–1889 (1994).

Reply to Referee A

This paper deals with gapless phase that appears at the phase transition region between the Stiefel-Whitney nontrivial ($C_{2z}T$ -symmetry-protected obstructed atomic) insulator and the trivial one. The Main finding of this work is that that this gapless becomes a critical metallic phase (CMP) under the random chemical potential with average $C_{2z}T$ symmetry. Furthermore, the authors show that the quantum-critical description of the CMP can be mapped into a percolation network model, with the regions featuring the trivial or unit absolute value Chern number, and the $C_{2z}T$ symmetry preserved on average. To numerically study this problem, the authors first consider a quantum network model on the Manhattan lattice. Subsequently, this model is regularized to an eight-band lattice model (to be able to explicitly feature the band topology) finding such a $C_{2z}T$ protected CMP and confirm the qualitative critical behavior as predicted from the previous network model.

We thank Referee A for carefully reading and summarizing our manuscript. The following concerns about general relevance help us improve our manuscript further. Next, we reply to the referee's criticism point by point.

The problem studied in the manuscript is of a very specialized nature, studying the behavior with the disorder of a phase with a very limited both theoretical and experimental relevance.

We thank the referee for this comment, which helped us improve the manuscript. In the revised manuscript, we generalize the discussion to generic magnetic point groups.

First, we show that the CMP survives even if $C_{2z}T$ is broken, yet m_{xy} and $C_{4z}T$ are respected on average. We added the relevant data to Sec. II.D of the main text. The following plot shows the (normalized) localization length Λ in the transition process in this setup:

For $\tilde{t} \in [1.2v/a, 1.7v/a]$, Λ does not decrease with the system size, which indicates the CMP. We also found CMP in a Kagome lattice model respecting $C_{6z}T$. (See the following figure where the red/blue triangles indicate the Chern blocks with Chern number $C = 1/-1$.) We added the relevant data to Sec. II.A of the supplementary material.

These results suggest that CMP may arise in transitions between OAI protected by other average symmetries, e.g., m_{xy} & $C_{4z}T$ and $C_{6z}T$, that forbid a net Hall conductance and protect the percolation mechanism.

To further show the generality, in Sec. II.D of the revised main text, we generalize the discussion of CMP to OAI transitions protected by simple (magnetic) groups Pm , $P2'$, $P4'$, and $P6'$. Instead of using the second Stiefel Whitney class (limited to $C_{2z}T$), we use the real space invariant (RSI) to characterize distinct OAIs for a given symmetry. RSI essentially reflects the inequivalent orbital occupations at the different Wyckoff positions. Hence, transitions between inequivalent OAIs will change the RSI. After a detailed analysis of RSIs (see Sec II.B of the supplementary material), we enumerated the OAI transitions with m , $C_{2z}T$, $C_{4z}T$, $C_{6z}T$ symmetries and summarized the results in the table on the next page.

One can see Sec. II.D for a detailed explanation of this table. In short, all the m , $C_{2z}T$, $C_{4z}T$, $C_{6z}T$ symmetries can support CMP alone under some conditions. We show that, first, the transitions between certain OAIs are accompanied by a finite gapless region with multiple linear or quadratic Dirac points; second, under certain types of disorders, these Dirac points give rise to a percolation picture in real space, leading to the CMP.

(Magnetic) Space groups & Wyckoff positions	Minimal OAI transitions	Minimal band structure transitions	Minimal Wannier center transitions
Pm-NSOC 1a [m] (x, 0) 1b [m] (x, 1/2)	$X_2 \rightarrow X_1$ $\begin{cases} \Delta_{A'}^a = 1 \text{ or } \Delta_{A''}^a = -1 \\ \Delta_{A'}^b = 1 \text{ or } \Delta_{A''}^b = -1 \end{cases}$		
	$M_2 \rightarrow M_1$ $\begin{cases} \Delta_{A'}^a = 1 \text{ or } \Delta_{A''}^a = -1 \\ \Delta_{A'}^b = -1 \text{ or } \Delta_{A''}^b = 1 \end{cases}$		
P2'-NSOC 1a [2'] (0, 0) 1b [2'] (1/2, 0) 1c [2'] (0, 1/2) 1d [2'] (1/2, 1/2)	NULL $\begin{cases} \Delta_{A'}^{a/b/c/d} = 0, \pm 1 \\ \sum_{f=a}^d \Delta_{A'}^f \neq 0 \end{cases}$		
P4'-NSOC 1a [4'] (0, 0) 1b [4'] (1/2, 1/2) 2c [2] (1/2, 0), (0, 1/2)	$\Gamma_1 \oplus \Gamma_1 \rightarrow \Gamma_2 \Gamma_2$ $\begin{cases} \Delta_B^c = -\Delta_A^a \\ = -\Delta_A^b = 1 \end{cases}$		
	$M_1 \oplus M_1 \rightarrow M_2 M_2$ $\begin{cases} \Delta_A^c = -\Delta_A^a \\ = -\Delta_A^b = 1 \end{cases}$		
	$X_2 \rightarrow X_1$ $\Delta_A^a = -\Delta_A^b = 1$		
P4'-SOC 1a [4'] (0, 0) 1b [4'] (1/2, 1/2) 2c [2] (1/2, 0), (0, 1/2)	$X_3 \rightarrow X_4$ $\begin{cases} \Delta_{1\bar{E}^2\bar{E}}^a = \pm 1 \text{ or } \Delta_{1\bar{E}^2\bar{E}}^b = \pm 1 \\ \Delta_{1E}^c = -1 \text{ or } \Delta_{1E}^d = 1 \end{cases}$		
	NULL $\Delta_{1E^2E}^a = -\Delta_{1E^2E}^b = 1$		
P6'-NSOC 1a [6'] (0, 0) 2b [3] (1/3, 2/3), (2/3, 1/3) 3c [2'] (1/2, 0), (0, 1/2), (1/2, 1/2)	$\Gamma_1 \oplus \Gamma_1 \rightarrow \Gamma_2 \Gamma_3$ $\Delta_{1E^2E}^a = -\Delta_{A_1}^b = 1$		
	$K_1 \oplus K_1 \rightarrow K_2 K_3$ $\Delta_{1E^2E}^a = -\Delta_{2E}^b = 1$		
	$K'_1 \oplus K'_1 \rightarrow K'_2 K'_3$ $\Delta_{1E^2E}^a = -\Delta_{1E}^b = 1$		
	NULL $\begin{cases} \Delta_{A_1}^a = \Delta_{1E^2E}^a \\ = -\Delta_{A_1}^c = \pm 1 \end{cases}$		

Furthermore, the model studied here, as also authors explicitly mention in the Discussion section, shares the same universality class as random Kane-Mele and Bernevig-Hughes-Zhang model, since both of them are with $C2zT$ symmetry but breaking time-reversal, which is the hallmark feature of the phase studied here (hence the title of the manuscript). This is expected from the universality hypothesis (symmetries, dimensionality and range of the disorder and/or interaction determine the universality class). Criticality of the disordered $C2zT$ KM and BHZ models has been found in Refs. 74, 75 and 48, 76, respectively, as also claimed by the authors in the Discussion section.

We thank the referee for this comment and are sorry for not properly explaining the relation between our work and previous literature. As the following paragraphs explain, our work fundamentally differs from previous works.

First, it is not clear to us how the CMP can be inferred from the universality hypothesis. In fact, CMP might be a counterexample of the commonly believed universality hypothesis. The only known criticality of Anderson transition in a 2D class A system is the Quantum Hall criticality, which has only a critical point, rather than a critical phase. The standard non-linear sigma model does not predict such a CMP in class A.

Second, Ref. 74 did not discuss CMP in the text at all. We cite this article because its data (Figs. 2 and 3) show a sign of criticality at a very small system size (up to $L=24$). We have clarified this in the revised manuscript. Ref. 48 is another work by the current authors and has not been posted yet. Note that Refs. 48, 74, 75 and 76 now become Refs. 58, 88, 89 and 90 in the revised main text.

Third, the CMP we found is also unpredictable from the results in Refs. 75 and 76. Both Refs. 75 and 76 ascribed the observed criticality to stable topology. They did not realize that CMP could happen between two completely trivial states, not to mention the change in Wannier centers. In particular, Ref. 75 claimed that the CMP is accompanied by the change of the spin Chern number, as the authors wrote in their abstract: “*The numerical simulations reveal sharp changes in the quantized values of C (spin Chern number) when crossing the regions of bulk extended states, indicating that the topological nature of the extended states is indeed linked to the spin-Chern numbers.*” Ref. 76 ascribed the criticality to two coupled quantum Hall transitions. The authors wrote in their abstract: “*We study the effects of random domains on the zero Hall plateau in QAH insulators ... the absence of structure inversion symmetry leads to a mixture between these two subsystems, gives rise to a line of critical points*”. Therefore, the processes they studied are topological phase transitions with additional weak perturbation terms, and the observed delocalizations are rooted in strong topology. In sharp contrast, our CMP is located between two completely trivial phases far away from any topological state. The only difference between the two trivial phases is in the Wannier centers. Since both the initial and final states are completely trivial and localized, at first glance, one would not expect delocalization in our lattice models. Therefore, our work - for the first time - underpins that CMP is a feature of the Wannier center transition and has nothing to do with strong topology. We have made this point clearer in the revised manuscript.

Fourth, Refs. 74-76 used system sizes up to $L=24, 32, 128$, respectively, which are much smaller than our system. In Sec. I.A and I.B of the revised supplementary materials, we verify the CMPs up to $L=2000$ for the network model and $L=500$ for the lattice model. Identifying the criticality requires a careful analysis of the scaling at large system sizes, as we will explain in the reply to the next question. Refs. 74-76 did not access a very large system size and did not present detailed scaling analyses as we have done.

Given the above, I find that this manuscript does not cut a high bar imposed by Nature Communications. It is more appropriate to a specialized journal, such as Scientific Reports in Nature Family, or Physical Review B.

As we have addressed the concerns about generality and novelty, we kindly ask the referee to reconsider her/his opinion of our work. First, as explained above, the CMP is not limited to $C_{2z}T$ symmetry. It can be generalized to generic magnetic point group symmetries such as $m, C_{4z}T, C_{6z}T$. Second, the CMP cannot be inferred from the previous works because the previously observed criticality is rooted in strong topology, and our model is far from stable topological states. Thus, our work is the first one to point out the relation between CMP and Wannier center transitions.

In my opinion, the readers would benefit tremendously if the supplementary materials have been organized more concisely.

We appreciate the kind suggestion of rearranging the supplementary material. In the revised supplementary material, we put the supplementary data in the first half for the convenience of reference. The technical details are set in the second half for readers interested in the detailed process and even reproducing the results.

Furthermore, I have the following technical points. To completely characterize the disorder at the transition between the CMP and localized phase, the authors should consider the following:

- 1. Calculate the critical exponents for the correlation length and the conductivity at this transition.*
- 2. At least discuss qualitatively, the appropriate continuum quantum field theory describing the transition from the CMP to the localized phase. In particular, find the critical exponents in this theory and compare with the ones obtained numerically.*

We are grateful for the two comments about the characteristics of the critical-localized transition. To address these suggestions, we did extensive numerical work on the network model, which greatly deepened our understanding of CMP.

As we have stressed, the transition differs from the traditional Anderson transition, where only a critical point is present. CMP, being a phase rather than a point, occupies a finite region of the parameter space (θ for the network model), which suggests a vanishing beta-function above some critical conductance that separates the localized and critical phases. We assume the numerical

beta-function at finite size L is $\beta(G, L) = \frac{L}{G} \cdot \frac{dG}{dL}$. It should approach the beta-function when $L \rightarrow \infty$. Applying the finite difference method, we obtain the beta-function of the Manhattan network:

We can see that β is linear on large conductance G when system size L is given, and the slope decreases as L increases. Also, notice that the system is a ballistic metal at the point $\theta = 0$, i.e., $G(\theta = 0, L) = G_0 L$, where G_0 is the conductance unit. Hence, for a given L , the beta-function must terminate at point $\beta(G_0 L, L) = 1$, which is confirmed by the data. Suppose the linear behavior of beta-function begins at a universal point $\beta(G_c, L) = 0$, then for $G \gg G_c$, the slope of the beta-function should be approximately $(G_0 L)^{-1}$ such that $\beta(G = G_0 L, L) = 1$. This conjecture implies $\beta(G, L) = \frac{G}{G_0 L}$ for large G . Solving this differential equation, we obtain

$$G(\theta, L) = \frac{1}{G_\infty(\theta)^{-1} + (G_0 L)^{-1}}$$

where $G_\infty(\theta) = G(\theta, L \rightarrow \infty)$. We found the above formula fits the conductance data very well for $G > 5G_0$, validating the conjecture. In the following $\ln G - \theta$ plot (the left hand side), the circles are the numerical results, and the curves are given by the above equation, where $G_\infty(\theta)$ is treated as a fitting parameter for each θ .

Further, we found a simple relation $G_\infty(\theta) \sim \theta^{-2}$, supported by the above plot on the right hand side.

According to the above scaling analysis, we are convinced that the $\beta(G, \infty)$ is exactly zero above some critical conductance $G_c = 2G_0 \sim 3G_0$. However, limited by the data precision, we cannot precisely determine G_c . Such a beta-function is non-analytic at G_c , which may suggest a two-parameter RG flow, such as the BKT transition.

The transition from the localized side is also worth investigating. We did finite size scaling inside the localized region but close to the transition boundary. We found the data precision of localization length is better than conductance in this region and, thus, is more suitable for finite size scaling.

Near the phase boundary, as long as the system size L is sufficiently large, the observables are determined by a single relevant parameter g , which characterizes the distance to the phase boundary. For a conventional phase transition where the correlation length in the thermodynamic limit is $\xi_\infty \sim g^{-\nu}$, one can prove that $\Lambda(g, L) \equiv \xi(g, L)/L = f(gL^{1/\nu})$, where $\xi(g, L)$ is the localization length with finite system size L , and f is a universal function. For a BKT-like transition where $\xi_\infty \sim \exp(ag^{-\nu})$, there is $\Lambda(g, L) = h(g(1 - a^{-1}g^\nu \ln L)^{-1/\nu})$, and h is also a universal function.

In our network model, the phase transition is controlled by a scattering angle θ . Hence, we can assume $g = \sum_{i=1}^m a_i(\theta - \theta_c)^i$. If one of these ansatzes is applicable, all the data points $\Lambda(\theta, L)$ should collapse to one curve in the $\Lambda - \phi$ plot where ϕ is either $gL^{1/\nu}$ or $g(1 - a^{-1}g^\nu \ln L)^{-1/\nu}$. Our data with $\theta \in (-1, -0.7)$ fit both ansatzes well.

For the ansatz $\phi = gL^{1/\nu}$, we found $\nu \approx 2.9$:

One parameter scaling, $\phi = gL^{1/\nu}$ and $\nu \approx 2.91$.

For ansatz $\phi = g(1 - a^{-1}g^\nu \ln L)^{-1/\nu}$, we found $\nu \approx 0.3 \sim 0.8$:

One parameter scaling, $\phi = g(1 - a^{-1}g^\nu \ln L)^{-1/\nu}$ and $\nu = 0.81$

Note that a standard BKT transition corresponds to $\nu = 0.5$.

Since both ansatzes work, we cannot exclusively decide the transition type from the localized side.

The above numerical analyses demonstrate a novel Anderson transition in class A that existing field theories cannot explain. At the current stage, we regard the effective BKT theory proposed by Shou-Cheng Zhang in [arXiv:cond-mat/9312010](https://arxiv.org/abs/cond-mat/9312010) as the most relevant field theory for the transition. However, this theory is controversial as other non-linear sigma model analyses by Mirlin [arXiv:cond-mat/9404070](https://arxiv.org/abs/cond-mat/9404070) and Efetov [arXiv:cond-mat/0010282](https://arxiv.org/abs/cond-mat/0010282) have reported conflicting results. This theory also essentially depends on a random magnetic field. Thus, it does not directly apply to our lattice models where only on-site disorders are present.

A complete scaling analysis and a correct effective field theory is challenging work. Hence, we leave it for future study. Nevertheless, we believe that our scaling analyses above have convincingly shown the novelty of the CMP.

Reply to Referee B

In this work, the authors propose and study a collection of related models in which an intervening critical metal phase is found as a C2T-protected obstructed atomic insulator is

tuned into another localized insulator by disorder which respects the C2T symmetry on average. The analysis began with a well-motivated network model on the Manhattan lattice, which is then regularized and truncated into two related lattice models. The realization of an OAI phase is verified from symmetry analysis. The presence of a critical metal phase as disorder is introduced is argued on the grounds of a percolation picture in terms of the network model, and is also evidenced by numerical results on both the network and lattice models.

We are grateful to Referee B for the careful reading and insightful comments that have improved our manuscript. We reply to all concerns and questions below.

Whether an intervening critical metal phase could exist in the disorder-induced transition between topologically trivial phases is an interesting problem, and I am convinced that the authors have provided a positive example involving an OAI on one side.

We appreciate the referee's positive evaluation for our work.

However, there is one question I believe the authors should address before I could make any recommendations: from the authors' general discussion, it was suggested that the intervening critical metal phase could be expected as a general phenomenon as the C2T-protected OAI is disorder-tuned into another localized phase. Yet, the models studied by the authors appeared to be more tailor-made than what would support the general claim. As the authors discussed the behavior of the network model could be understood from the percolation problem of the random flux model. Since the lattice models descended from the network model, it is unclear if they are sufficiently general to reveal the generic behavior of how the C2T-protected OAI responds to on-site potential disorder.

We thank the referee for this insightful comment and are sorry for the confusion we have caused. In fact, we do not claim that CMP necessarily appears in a transition between two inequivalent $C_{2z}T$ -protected OAIs. For example, in Fig. 2(g) of the manuscript, one can find a localized path (in the white region) connecting the two OAIs separated in the clean limit. Thus, whether CMP appears should depend on the Hamiltonian parameters. For another example, we consider a variant of the Hamiltonian in Eq. (4), where ϕ_{pq} , the only parameter that breaks time-reversal symmetry T , is set to zero. When $\phi_{pq} = 0$, the $C_{2z}T$ symmetry is enhanced to C_{2z} and T . The two OAIs are still inequivalent under these symmetries (including original $C_{2z}T$). However, as the model is in class AI, the OAI transition will be localized under any finite disorder strength due to the weak localization effect. We give more details about this calculation below, in our response to referee C.

On the other hand, this makes CMP more interesting: either disorder strength W or ϕ_{pq} can realize a continuous crossover from a localized state to CMP. As detailed in the reply to Referee A, careful scaling analysis suggests that this crossover is likely a BKT-type phase transition. However, our current data cannot fully determine this and we leave it for future studies. In the revised manuscript, we add the ϕ_{pq} dependence of localization length.

In regard to generality, we can say that OAI transitions without time-reversal symmetry provide a generic mechanism to realize CMP. In addition, the OAI transition is not limited to $C_{2z}T$ but also

applies to generic magnetic symmetries. We have explained this in the response to Referee A, and we repeat it here:

We found that the CMP survives even if $C_{2z}T$ is broken, yet m_{xy} and $C_{4z}T$ are respected on average. We added the relevant data to Sec. II.D of the main text. The following plot shows the (normalized) localization length Λ in the transition process in this setup:

We also found CMP in a Kagome lattice model respecting $C_{6z}T$. We added the relevant data to Sec. II.A of the supplementary material.

These results suggest that CMP may arise in transitions between OAI protected by other average symmetries, e.g., m_{xy} & $C_{4z}T$ and $C_{6z}T$, that forbid a net Hall conductance and protect the percolation mechanism.

To further show the generality, in Sec. II.D of the revised main text, we generalize the discussion of CMP to OAI transitions protected by simple (magnetic) groups Pm , $P2'$, $P4'$, and $P6'$. Instead of using the second Stiefel Whitney class (limited to $C_{2z}T$), we use the real space invariant (RSI) to characterize distinct OAI for a given symmetry. RSI essentially reflects the inequivalent orbital

occupations at the Wyckoff positions. Hence, transitions between inequivalent OAI will change RSIs. After a detailed analysis of RSIs (see Sec II.B of the supplementary material), we enumerated the OAI transitions with m , $C_{2z}T$, $C_{4z}T$, $C_{6z}T$ symmetries and summarized the results in the table on the next page.

One can see Sec. II.D for a detailed explanation of this table. In short, all the m , $C_{2z}T$, $C_{4z}T$, $C_{6z}T$ symmetries can support CMP alone under some conditions. We show that, first, the transitions between certain OAIs are accompanied by a finite gapless region with multiple linear or quadratic Dirac points; second, under certain types of disorders, these Dirac points give rise to a percolation picture in real space, leading to the CMP.

For instance, while it is neat that the random vector potential disorder in the network model could be mapped to potential disorder (by dropping the smaller off-diagonal terms and the sub-leading hopping disorder), the potential disorder one obtains at the end appears to be locally correlated. More specifically, from Eqs. (B41)-(B45) in the supplementary information, it appears that the vector potential disorder A , which are independent on different edges, ultimately enters into the on-site disorder term of the various corner orbitals in a unit cell through a very specific combination.

We thank the referee for the careful reading and the insight about disorder correlations. At the early stage of this project, we found a CMP with full disorder (without dropping off-diagonal terms and correlations). The data also shows a CMP:

(Magnetic) Space groups & Wyckoff positions	Minimal OAI transitions	Minimal band structure transitions	Minimal Wannier center transitions
Pm-NSOC 1a [m] (x, 0) 1b [m] (x, 1/2)	$X_2 \rightarrow X_1$ $\begin{cases} \Delta_{A'}^a = 1 \text{ or } \Delta_{A''}^a = -1 \\ \Delta_{A'}^b = 1 \text{ or } \Delta_{A''}^b = -1 \end{cases}$		
	$M_2 \rightarrow M_1$ $\begin{cases} \Delta_{A'}^a = 1 \text{ or } \Delta_{A''}^a = -1 \\ \Delta_{A'}^b = -1 \text{ or } \Delta_{A''}^b = 1 \end{cases}$		
P2'-NSOC 1a [2'] (0, 0) 1b [2'] (1/2, 0) 1c [2'] (0, 1/2) 1d [2'] (1/2, 1/2)	NULL $\begin{cases} \Delta_A^{a/b/c/d} = 0, \pm 1 \\ \sum_{j=a}^d \Delta_A^j \neq 0 \end{cases}$		
P4'-NSOC 1a [4'] (0, 0) 1b [4'] (1/2, 1/2) 2c [2] (1/2, 0), (0, 1/2)	$\Gamma_1 \oplus \Gamma_1 \rightarrow \Gamma_2 \Gamma_2$ $\begin{cases} \Delta_B^c = -\Delta_A^a \\ = -\Delta_A^b = 1 \end{cases}$		
	$M_1 \oplus M_1 \rightarrow M_2 M_2$ $\begin{cases} \Delta_A^c = -\Delta_A^a \\ = -\Delta_A^b = 1 \end{cases}$		
	$X_2 \rightarrow X_1$ $\Delta_A^a = -\Delta_A^b = 1$		
P4'-SOC 1a [4'] (0, 0) 1b [4'] (1/2, 1/2) 2c [2] (1/2, 0), (0, 1/2)	$X_3 \rightarrow X_4$ $\begin{cases} \Delta_{1E^2E}^a = \pm 1 \text{ or } \Delta_{1E^2E}^b = \pm 1 \\ \Delta_{1E}^c = -1 \text{ or } \Delta_{1E}^d = 1 \end{cases}$		
	NULL $\Delta_{1E^2E}^a = -\Delta_{1E^2E}^b = 1$		
P6'-NSOC 1a [6'] (0, 0) 2b [3] (1/3, 2/3), (2/3, 1/3) 3c [2'] (1/2, 0), (0, 1/2), (1/2, 1/2)	$\Gamma_1 \oplus \Gamma_1 \rightarrow \Gamma_2 \Gamma_3$ $\Delta_{1E^2E}^a = -\Delta_{A_1}^b = 1$		
	$K_1 \oplus K_1 \rightarrow K_2 K_3$ $\Delta_{1E^2E}^a = -\Delta_{2E}^b = 1$		
	$K_1' \oplus K_1' \rightarrow K_2' K_3'$ $\Delta_{1E^2E}^a = -\Delta_{1E}^b = 1$		
	NULL $\begin{cases} \Delta_{A_1}^a = \Delta_{1E^2E}^a \\ = -\Delta_A^c = \pm 1 \end{cases}$		

Later on, we tried uncorrelated on-site disorder and found no qualitative change in CMP. For simplicity and numerical efficiency, we have used uncorrelated on-site disorder ever since. We have clarified the dropping of the correlation in the revised manuscript.

If the authors' claim is indeed general, then any on-site disorder which respects C2T on average which tunes the OAI into the targeted localized phase should similarly induce the intervening critical metal phase. Is this true?

We thank the referee for this question. As we have explained in reply to the first question, we do not expect CMP to necessarily exist for arbitrary Hamiltonian. We have shown that tuning parameters in the clean Hamiltonian can continuously turn off the CMP.

Alternatively, the authors may explain why the perceived local correlation in the potential disorder obtained is absent and there is no correlation beyond requiring the disorder ensemble to respect C2T on average.

We thank the referee for this question. As we have explained in the reply to the second question, the disorder correlation is not relevant to the existence of CMP.

Lastly, a minor comment: in writing $E_n(k) = E_{\{n+\delta\}}(k)$ above Eq. (3) of the main text, it may be clearer to mention the shift in energy/ repeating window.

We thank the referee for this kind suggestion. We have fixed this mistake in the revised manuscript.

Reply to Referee C

The past decade has seen significant advancements in understanding of topological phases protected by crystalline symmetries. Recently, intriguing characteristics of topological crystalline phases have been discovered. Along this line, the authors presented evidence for a critical metallic phase between an obstructed atomic insulator and a genuine trivial one.

Although I found the presented manuscript interesting, I have concerns about the novelty of the findings presented in the manuscript. More specifically, I wonder if experts in the field might easily anticipate the critical metallic phase in this work. Furthermore, I am concerned that the presented results are not as impactful as the journal expects. This expectation is based on the following three things:

We thank Referee C for careful reading and helpful comments. In the following, we provide a detailed response to the questions and explain that the CMP cannot be easily anticipated (by giving a counterexample of anticipation according to K-theory). We hope the referee can reconsider her/his opinion about the novelty of our work.

(i) As the authors mentioned in Introduction, Ref. 15 has established a connection between localization/delocalization and the classification of massive/massless Dirac Hamiltonians (equivalently, K-theory classification). Also, Ref. 15 has extended the discussion to the case with reflection symmetry. Furthermore, the classification of Dirac Hamiltonians in the presence of point group symmetries has been established in [arxiv:1811.01977](https://arxiv.org/abs/1811.01977) and [1907.09354](https://arxiv.org/abs/1907.09354). Importantly, K-groups often contain atomic insulators whose electrons are localized at some points in real space.

We thank the referee for asking this valuable question. Indeed, Ref. 15 established a connection between delocalization and the Dirac Hamiltonian (K-theory classification). However, the discussion is limited to stable topological states with robust gapless boundary states. The mirror topological crystalline insulator discussed in Ref. 15 also belongs to the stable topological state. The common belief is that whenever the bulk topological state has a robust boundary state, its transition to a trivial insulator must undergo delocalization in the bulk, such that boundary states on two opposite boundaries may mutually annihilate. This can be understood intuitively: Suppose the disorder potential slowly varies in real space. Then, during the transition, the disordered system can be divided into topological and trivial regions. Boundary states between the two types of regions must exist as promised by stable topology, giving rise to the delocalized phase transition.

We also thank the referee for informing us of the two important works, [arXiv:1811.01977](https://arxiv.org/abs/1811.01977) and [arXiv:1907.09354](https://arxiv.org/abs/1907.09354), which use K-theory to classify topological states protected by (magnetic) point group symmetries. We believe that these theories will be beneficial in proving the generic statement that phase transitions changing (crystalline-symmetry protected) stable topology must undergo delocalizations. We have added citations to the two works in the revised manuscript.

However, this picture or argument does not apply to transitions between OAI states because there is no guaranteed boundary state between inequivalent OAIs. We provide a concrete counterexample in the response to the next question, where a transition between distinct OAIs is proven localized.

(ii) According to recent classification results, the K-group is $(Z_2)^3 \times Z$ in magnetic layer group $p112'$, where all generators possess localized electrons. In this symmetry setting, there are four inequivalent Wyckoff positions: $(x,y) = (0,0), (1/2, 0), (0, 1/2),$ and $(1/2, 1/2)$. In fact, roughly speaking, generators of $(Z_2)^3$ correspond to the "difference" between two insulators with localized electrons at two different Wyckoff positions. In this case, we can say that nontrivial elements of $(Z_2)^3$ are obstructed atomic insulators with some appropriate choice of a reference state in K-theory.

(iii) In Ref. 39, some of the authors have revealed that such a metallic phase exists between a trivial insulator and a nontrivial element of the K-group (an axion insulator) when inversion symmetry is respected on average.

The first two points imply that the emergence of such critical metallic phases is a common feature of phase transitions between two different elements of a K-group.

We thank the referee for this meaningful comment. Here, we give a counterexample illustrating how the transition between (K-theory) distinct OAIs can be localized. We consider the Hamiltonian in Eq. (4) of the manuscript:

$$H_{8B} = \tilde{t} \sum_{\langle p,q \rangle} c_p^\dagger c_q + t \sum_{\langle\langle p,q \rangle\rangle} e^{i\phi_{pq}} c_p^\dagger c_q + t' \sum_{\langle\langle\langle p,q \rangle\rangle\rangle} c_p^\dagger c_q$$

where \tilde{t} , t , and t' are all real parameters representing the hoppings on the green, orange, and dashed bonds, respectively. $\phi_{pq} = 3\pi/4$ ($-3\pi/4$) if the associated hopping is parallel (anti-parallel) to the orange arrows. When $\tilde{t} = 0$, the model is in an OAI limit where electrons are located at the C4 centers (the plaquets). When $\tilde{t} \rightarrow \infty$, the model is the other OAI limit where electrons are located at the C2T centers (green bonds). Thus, tuning \tilde{t} realizes a phase transition between the two OAIs.

One should notice that ϕ_{pq} is the only parameter that breaks time-reversal symmetry. We then consider a model with $|\phi_{pq}| = 0$ or π . The layer group $p112'$ is enhanced to the gray group $p1121'$. Here, we ignore other symmetries in H_{8B} , which will not affect the following discussion. According to the real space invariant (RSI) theory in Ref. 55 (which becomes Ref. 65 in the revised manuscript), each C_{2z} -invariant Wyckoff position now has a Z-valued invariant $\delta = m_+ - m_-$, where m_{\pm} is the number of occupied even (odd) Wannier functions at this Wyckoff position. (δ

reduces to a Z2 quantity if C_{2z} and T are broken, yet $C_{2z}T$ is respected.) A K-theory analysis of $p1121'$ will give the same invariant. Now, at the $\phi_{pq} = 0$ or π limit, tuning \tilde{t} from 0 ($\delta = 0$) to infinity ($\delta = -1$) still drives a phase transition between distinct OAs. However, the phase transition must be localized due to the weak localization effect in the presence of spinless time-reversal symmetry.

To demonstrate localization, we start from the most delocalized point in the original model, i.e., the curve top ($\tilde{t} = \frac{\pi v}{2a}$) in Fig. 2(f), and continuously turn off TRS-breaking. To be specific, we set $t = \frac{\sqrt{2}\pi v}{4a}$, $t' = -\frac{\pi v}{4a}$, $\tilde{t} = \frac{\pi v}{2a}$, $W = \frac{3v}{a}$, $E_F = 0$. We plot the normalized localization length as a function of $|\phi_{pq}|$:

One can see that the system become localized near $|\phi_{pq}| = \pi$. For all the scanned $|\phi_{pq}|$, the band structure in the clean limit is gapless near the Fermi level, i.e., the system stays at the intermediate state of the OAI transition. The localized phase will shrink with weaker disorder, yet the point $|\phi_{pq}| = \pi$ is always localized. Hence, the OAI transition with $|\phi_{pq}| = \pi$ is localized.

Furthermore, the third point has already pointed out the localization/delocalization protected by crystalline symmetries on average.

The axion insulator is a stable topological state and possesses robust gapless boundary states. Thus, its transition to a trivial insulator is expected to be delocalized from the domain wall picture explained in the reply to the first question.

I respectfully ask the authors to clarify whether the critical metric phase in this work is completely different from those among distinct elements of K-groups.

We thank the referee for this comment. Using the above counterexample, we have convincingly shown that a transition between distinct K-group elements can be localized if the two elements are equivalent under stable-equivalence, e.g., distinct OAs. The mechanism of well-known delocalization transitions between stable topological states can be understood in the domain wall

picture explained in the reply to the first question. This mechanism does not apply to the OAI transitions we consider, and we show a counterexample where an OAI transition is localized. Therefore, the CMP in this work is indeed completely different from those in previous works.

On the side of the presentation, I would like to ask the authors to consider the following comments/questions.

I could not find the definition of “respecting symmetries on average.” Since this is a key concept of this work, the authors should add a precise definition somewhere.

We appreciate this kind suggestion that helps us clarify the critical concept of average symmetry. According to [arXiv:2209.02723](https://arxiv.org/abs/2209.02723), average symmetry is a symmetry in an ensemble that comprises different disorder realizations on a local Hamiltonian. The traditional symmetry transforms an individual system into its own, and this type of symmetry is now known as the exact symmetry. Average symmetry transforms an individual system into another with the same realization probability. Moreover, we require that any two systems in the ensemble can adiabatically deform to each other without breaking the exact symmetries. Our ensemble satisfies the definition, and we added it to the revised manuscript.

Considering the current length of the manuscript and the existence of Method section, I recommend them to move some technical details from Supplementary Information to Method or main text. For example, in Figures 1 and 2, the authors use symbols of irreducible representations. I think, although the manuscript is well written, the manuscript would be more readable when Tables I and II are included in Method.

We thank the referee for this helpful comment. We have added the Methods section and moved tables I and II to the Methods section.

REVIEWERS' COMMENTS

Reviewer #2 (Remarks to the Author):

I have reviewed the initial round of reports from the other referees, as well as the authors' responses to them. I sincerely appreciate the authors for their significant effort in revising the manuscript and providing a comprehensive explanation that clarifies the physics. Additionally, I apologize for any confusion caused by the word usage in my previous report. As the authors correctly pointed out, they never claimed that a critical metal phase is inevitable in the disorder-induced transition between the two distinct obstructed atomic insulators (OAI). Instead, I was referring to the general existence of a critical metal phase in the phase diagram, not the necessity of passing through it along any possible paths.

As noted by all the referees, while the manuscript is of interest to experts in the field, there are concerns regarding its general applicability and appeal to a broader readership. Essentially, the authors have convincingly demonstrated that an intervening critical metal phase could arise in the disorder-induced transition between the two OAI within the models they studied. However, it remains unclear whether the same phase will appear in other models that may be more relevant to materials or other experimental realizations. The general problem studied, namely how disorder could impact the phase transition between topologically distinct phases, is also relatively old. Although I acknowledged the authors' contribution in identifying the possibility of an intervening critical metal phase between two topologically trivial but mutually distinct insulators, the results may be more suitable for a more specialized journal than *Nature Communications*.

Reviewer #3 (Remarks to the Author):

I would like to thank the authors for correcting my misunderstanding. I now understand that my naive expectations were incorrect and that the results obtained are indeed highly nontrivial. The authors' extensive efforts have greatly improved the manuscript, in particular by including discussions on various magnetic point groups, additional numerical data, and clarifications on the differences in CMP phases between this work and existing literature. This has significantly improved the quality of the manuscript.

Regarding general interest, as the authors have noted, the concept of average symmetry has been recently recast in Phys. Rev. X 13, 031016. Since this work is in line with this development, I believe that publication of this manuscript would be valuable to a broad readership of *Nature Communications*. Therefore, I recommend the publication of this manuscript in its current form.

In the following we provide a point-by-point response to the reviewers' comments.

Reviewer #2

Reviewer: I have reviewed the initial round of reports from the other referees, as well as the authors' responses to them. I sincerely appreciate the authors for their significant effort in revising the manuscript and providing a comprehensive explanation that clarifies the physics. Additionally, I apologize for any confusion caused by the word usage in my previous report. As the authors correctly pointed out, they never claimed that a critical metal phase is inevitable in the disorder-induced transition between the two distinct obstructed atomic insulators (OAI). Instead, I was referring to the general existence of a critical metal phase in the phase diagram, not the necessity of passing through it along any possible paths.

Reply: We thank the reviewer for the input in reviewing our manuscript. Our manuscript has benefitted a lot from the constructive comments of the reviewer.

Reviewer: As noted by all the referees, while the manuscript is of interest to experts in the field, there are concerns regarding its general applicability and appeal to a broader readership. Essentially, the authors have convincingly demonstrated that an intervening critical metal phase could arise in the disorder-induced transition between the two OAI within the models they studied. However, it remains unclear whether the same phase will appear in other models that may be more relevant to materials or other experimental realizations.

Reply: In addition to numerical calculations about our various models, which descend from the Manhattan network model and show the criticality, we also analyzed the existence of CMP in generic models:

- (i) We provide a Dirac theory argument, where the Dirac points are protected by an average symmetry that is broken by the disorder. The Dirac theory reveals that a large number of Dirac points generally favor the CMP.
- (ii) We systematically analyzed the phase transitions between OAIs protected by magnetic space groups P_m , $P2'$, $P4'$, $P6'$ with and without spin-orbit couplings. See Fig. 6. For each group, we have enumerated all the minimal phase transitions between OAIs; For each minimal transition, we have determined the number and type of Dirac points in the intermediate phase. For example, the transition of charge centers between 1a and 3c positions in $P6'$ will at least involve six Dirac points, which, according to the Dirac theory in the last paragraph, can support a CMP. Our square lattice models are also examples of generic OAI transitions.
- (iii) We also provide a Kagome lattice model that exhibits CMP.

Based on the first two points, we have provided rules to find CMP in generic magnetic space groups, i.e., P_m , $P2'$, $P4'$, $P6'$, and **all** their parent magnetic space groups. Thus, we believe we have addressed the issue of general applicability.

Predicting CMP in realistic materials is a challenging task and beyond the scope of the current study. We leave it for future studies.

As our work builds a new connection between topology (trivial OAI states) and delocalization, it may interest readers on both sides. Also, as pointed out by reviewer #3, the average-symmetry-protected topological state is a new concept that has attracted increasing attention. Our work demonstrates the potential application of the average symmetry in delocalized states. Thus, it may be valuable to a broader readership.

The general problem studied, namely how disorder could impact the phase transition between topologically distinct phases, is also relatively old.

Reply: We respectfully disagree with the reviewer at this point. As we have replied to reviewer #3 in the last round, the old studies are mostly about the transitions between stable topological states. Very rare studies have been conducted on transitions between trivial states. The two are fundamentally different. Therefore, we are not convinced that the subject of our work is relatively old.

We repeat part of the response to reviewer #3 in the last round here: *“The common belief is that whenever the bulk topological state has a robust boundary state, its transition to a trivial insulator must undergo delocalization in the bulk, such that boundary states on two opposite boundaries may mutually annihilate. This can be understood intuitively: Suppose the disorder potential slowly varies in real space. Then, during the transition, the disordered system can be divided into topological and trivial regions. Boundary states between the two types of regions must exist as promised by stable topology, giving rise to the delocalized phase transition... However, this picture or argument does not apply to transitions between OAI states because there is no guaranteed boundary state between inequivalent OAIs. We provide a concrete counterexample in the response to the next question, where a transition between distinct OAIs is proven localized.”*

Although I acknowledged the authors' contribution in identifying the possibility of an intervening critical metal phase between two topologically trivial but mutually distinct insulators, the results may be more suitable for a more specialized journal than Nature Communications.

Reviewer #3

I would like to thank the authors for correcting my misunderstanding. I now understand that my naive expectations were incorrect and that the results obtained are indeed highly nontrivial. The authors' extensive efforts have greatly improved the manuscript, in particular by including discussions on various magnetic point groups, additional

numerical data, and clarifications on the differences in CMP phases between this work and existing literature. This has significantly improved the quality of the manuscript.

Regarding general interest, as the authors have noted, the concept of average symmetry has been recently recast in Phys. Rev. X 13, 031016. Since this work is in line with this development, I believe that publication of this manuscript would be valuable to a broad readership of Nature Communications. Therefore, I recommend the publication of this manuscript in its current form.

Reply: We thank reviewer #3 for the positive evaluation of our work and the recommendation for publication. We also appreciate the helpful comments in the previous round that have helped us improve our manuscript.